

# Radiocarbon dating of glacier ice

Chiara Uglietti[1,2,3], Alexander Zapf[1,2,3,†], Theo M. Jenk[1,3], Sönke Szidat[2,3], Gary Salazar[2], Margit Schwikowski[1,2,3]

[1]Laboratory of Environmental Chemistry, Paul Scherrer Institute, 5232 Villigen PSI, Switzerland
[2]Department of Chemistry and Biochemistry, University of Bern, 3012 Bern, Switzerland
[3]Oeschger Centre for Climate Change Research, University of Bern, 3012 Bern, Switzerland
†deceased

*Correspondence to*: theo.jenk@psi.ch

**Abstract.** High altitude glaciers and ice caps from mid-latitudes and tropical regions contain valuable signals of past climatic and environmental conditions as well as human activities, but for a meaningful interpretation this information needs to be placed in a precise chronological context. For dating the upper part of ice cores from such sites several relatively precise methods exist, but they fail in the older and deeper part, where plastic deformation of the ice results in strong annual layer thinning and a non-linear age-depth relationship. If sufficient organic matter such as plant, wood or insect fragments were found, radiocarbon ($^{14}$C) analysis had thus been the only option for a direct and absolute dating of deeper ice core sections. However such fragments are rarely found and even then very likely not at the depths and in the resolution desired. About 10 years ago, a new, complementary dating tool was therefore introduced by our group. It is based on extracting the µg-amounts of the water-insoluble organic carbon (WIOC) fraction of carbonaceous aerosols embedded in the ice matrix for subsequent $^{14}$C dating. Meanwhile this new approach was improved considerably, thereby reducing the measurement time and improving the overall precision. Samples with ~10 µg WIOC mass can now be dated with reasonable uncertainty of around 10-20% (variable depending on sample age). This requires about 100 to 500 g of ice considering the WIOC concentrations typically found in mid- and low-latitude glacier ice. Dating polar ice with satisfactory age precision is still not possible since WIOC concentrations are around one order of magnitude lower. The accuracy of the $^{14}$C WIOC method was validated by applying it to independently dated ice. With this method the deepest parts of the ice cores from Colle Gnifetti and Mt. Ortles glacier in the European Alps, Illimani glacier in the Bolivian Andes, Tsambagarav ice cap in the Mongolian Altai, and Belukha glacier in the Siberian Altai have been dated. In all cases a strong annual layer thinning towards bedrock was observed and the oldest ages obtained were in the range of 10000 yrs. $^{14}$C WIOC-dating was not only crucial for interpretation of the embedded environmental and climatic histories, but additionally gave a better insight into glacier flow dynamics close to bedrock and past glacier coverage. For this the availability of multiple dating points in the deepest parts was essential, which is the strength of the presented WIOC $^{14}$C-dating method, allowing determination of absolute ages from principally every piece of ice.

**Keywords:** ice cores, mid- and low latitude glaciers, water-insoluble organic carbon, radiocarbon, chronology

## 1 Introduction

High altitude glaciers and ice caps from mid-latitudes and tropical regions contain valuable signals of past climate and atmospheric variability at regional and local scale and are located in areas with large biological diversity and inhabited by the majority of the world's population. Particularly mid-latitudes glaciers, for instance in the European Alps or in the Himalaya, are influenced by the nearby anthropogenic pollution sources, thereby additionally preserving the signature of human activities. This information can generally be retrieved from glacier ice cores, but needs to be placed in a precise chronological context to allow meaningful interpretation with respect to environmental and climatic changes.

Ice core dating is a sophisticated task and the most common approach is annual layer counting, which relies on seasonally fluctuating signals. A number of ice core parameters such as the stable isotope ratio of hydrogen or oxygen in the water ($\delta^2$H,



$\delta^{18}$O), the concentration of trace components (e.g. ammonium, mineral-dust-related trace elements, black carbon), and the presence of melt layers may vary with the seasons. To reduce uncertainty in layer counting the time scale is additionally anchored with reference horizons like the radioactivity peak resulting from nuclear weapon tests in the 1960s or tephra and aerosol layers caused by volcanic eruptions (Thompson et al., 1998; Schwikowski, 2004; Eichler et al., 2009; Moore et al.,

2012; Thompson et al., 2013). An independent method is nuclear dating with the naturally occurring radioisotope $^{210}$Pb. Determined by the $^{210}$Pb half-life of 22.3 years and its atmospheric concentration, the time period accessible for dating is in the order of a century (Gäggeler et al., 1983; Eichler et al., 2000; Herren et al., 2013). All these dating techniques fail in the older and deeper part of glaciers, where plastic deformation of the ice, under the weight of the overlying mass, results in horizontal ice flow, stretching annual layers continuously with increasing depth. Correspondingly, the depth-age relationship

of high-alpine glaciers is strongly non-linear (Jenk et al., 2009) and annual layers and also volcanic signals become undetectable below a certain depth with the current spatial resolution of most analytical methods. Glacier flow modelling can only give rough age estimates with large uncertainties close to the bedrock of high-alpine glaciers (Lüthi and Funk, 2001). Radiocarbon ($^{14}$C) analysis has been the only option allowing a direct and absolute dating of these deeper ice core sections in the rare cases when sufficient organic matter such as plant, wood or insect fragments were found (Thompson et al., 1998;

Thompson et al., 2002). However, in glacier ice such findings do not only happen very seldomly but even if lucky, they do not allow for continuous or at least regular dating which limits not only the application of the $^{14}$C technique but also its use to derive a complete chronology based on absolutely dated layers.

A new, complementary dating tool was therefore introduced by our group about 10 years ago, which is based on extracting the µg-amounts of the water-insoluble organic carbon fraction of carbonaceous aerosols embedded in the ice matrix for $^{14}$C

dating (Jenk et al., 2006; Jenk et al., 2007). Carbonaceous compounds represent a large, but highly variable fraction of the atmospheric aerosol mass (Gelencsér, 2004; Hallquist et al., 2009). Total organic carbon (TOC, also referred to as total carbon, TC) is instrumentally divided into two sub-fractions according to their refractory and optical properties. Elemental carbon (EC) consists of highly polymerized substances which are extremely refractory and light absorbent and therefore this fraction is also called black carbon (BC) or soot (Gelencsér, 2004; Hallquist et al., 2009). EC derives merely from the

incomplete combustion of fossil fuels and biomass. Organic carbon (OC) is formed by weakly refractory hydrocarbons of low to medium molecular weight. Whereas EC is generally insoluble in water, OC is further subdivided into water-soluble organic carbon (WSOC) and water-insoluble organic carbon (WIOC) (Szidat et al., 2004a). In water samples the former is also known as dissolved organic carbon (DOC) (Legrand et al., 2013; May et al., 2013). OC is emitted directly as primary aerosol from a vast diversity of sources and emission processes, including mobilization of plant debris, pollen, vegetation

waxes, microorganisms, spores, the organic fraction of soil as well as emissions from biomass burning (e.g. forest fires) and anthropogenic processes (biomass burning and fossil fuel combustion), but it is also formed in the atmosphere by oxidation of gaseous precursors as secondary organic aerosol (Gelencsér, 2004; Gelencsér et al., 2007; Hallquist et al., 2009).

Carbonaceous aerosols are transported in the atmosphere to high-alpine glaciers, where they may be deposited by both wet and dry deposition processes and finally embedded in glacier ice (Lavanchy et al., 1999; Jenk et al., 2006; Legrand and

Puxbaum, 2007; McConnell et al., 2007; Kaspari et al., 2011). Consequently using carbonaceous aerosols allows dating any piece of ice core, given that it contains sufficient carbon mass. The WSOC fraction (i.e. DOC) would be ideal for dating, since it has the highest concentrations in ice. However, its extraction is complicated. It involves the outgassing of aqueous atmospheric $CO_2$, removal of dissolved carbonates, wet oxidation of the organic compounds to $CO_2$ under inert gas, and finally quantitative trapping of the evolved $CO_2$ (May et al., 2013). Since major contributors of DOC, like light carboxylic

acids, are ubiquitous in the air, all these steps are prone to contamination. Therefore from the different carbonaceous particle fractions we selected WIOC as target for $^{14}$C dating for several reasons. First, WIOC is mainly of biogenic origin in pre-industrial times (Jenk et al., 2006) and therefore supposed to contain a contemporary $^{14}$C signal representative of the age of the ice (Jenk et al., 2006; Steier et al., 2006). Second, the average WIOC concentration in ice is higher than the respective



EC concentration, allowing for smaller ice samples and potentially higher time resolution, which consequently provides a

better signal to noise ratio (mainly determined by the overall blank) and smaller uncertainty of the dating results. Third, OC has a lower probability compared to EC for in-built reservoir ages from e.g. burning of old trees or old organic matter. Moreover OC is insensitive to potentially insufficiently removed carbonates in mineral dust rich layers (e.g. Saharan dust), which may contribute to the EC fraction because of the higher combustion temperature applied to EC (Jenk et al., 2006). The extraction of WIOC from the ice is straightforward as it can be collected by filtration of the melted ice. Note that in previous

publications (Sigl et al., 2009; Zapf et al., 2013) the term POC was used for particulate organic carbon (Drosg et al., 2007). Since POC can be mistaken with primary organic carbon (Gelencsér, 2004; Zhang et al., 2012) we adopted the term water-insoluble organic carbon (WIOC) instead in this overview.

Our research group has a long history in [14]C dating of ice cores using the aforementioned WIOC fraction of carbonaceous particles. Lavanchy et al. (1999) introduced initial methods to determine the concentrations of carbonaceous particles in ice

from a European high-alpine glacier. Next, the methodology was developed for source apportionment of aerosols by [14]C measurements in different carbonaceous particle fractions (Szidat et al., 2004b). This was conducted in close collaboration with the Laboratory of Ion Beam Physics of the ETH Zurich, a well established [14]C dating facility and a world-leading group in Accelerator Mass Spectrometry (AMS) technology, where simultaneously and continuously the analytical aspect of instrumentation was improved (Synal et al., 2000; Ruff et al., 2007; Synal et al., 2007; Ruff et al., 2010). The methodology

of [14]C analysis of the different carbonaceous particle fractions was adopted to study the suitability of WIOC for [14]C dating of old ice, finding that it is of purely biogenic origin prior to industrialization (Jenk et al., 2006; Jenk et al., 2007). Since then this novel [14]C approach has been applied for dating a number of ice cores from different high-altitude mountain glaciers (Table 1), (Jenk et al., 2009; Sigl et al., 2009; Kellerhals et al., 2010; Herren et al., 2013; Zapf et al., 2013; Aizen et al., 2016). Meanwhile the method has been further optimized and was additionally validated by determining the age of

independently dated ice. Here we give an overview of the current status of the now routinely applied [14]C dating method for glacier ice by presenting an update on recent optimizations and discussing the potential of this novel approach.

## 2 Sample preparation, OC/EC separation and [14]C analysis

The preparation of ice samples follows the procedure according to Jenk et al. (2007). First, samples are decontaminated in a cold room (-20˚C) by removing the outer layer (3 mm) with a pre-cleaned stainless steel band saw, followed by rinsing the

samples with ultra-pure water (18 MΩ cm quality) in a class 100 clean bench. Around 20-30% of the ice samples' mass is lost during these first steps, resulting in a final mass of about 100 to 500 g (initial mass of at least around 600-800 g of ice). The samples are then transferred and stored frozen at -20˚C in pre-cleaned (soaked and rinsed for three days with daily exchanged ultra-pure water) 1-L-containers (Semadeni, PETG) until being melted at room temperature directly before filtration. To ensure that carbonates potentially present in the ice are completely dissolved, ~20 mL of 1M HCl (30%

Suprapure, Merck) are added to the melted samples (Cao et al., 2013), resulting in a pH of < 2, before being sonicated for 5 min. Subsequently, the insoluble carbonaceous particles are filtered onto preheated (5h at 800˚C) quartz fibre filters (Pallflex Tissuquartz, 2500QAO-UP), using a dedicated glass filtration unit, also carefully pre-cleaned by rinsing with ultra-pure water and by baking the glass at 450˚C for 3h. As a second carbonate removal step, the filters are acidified three times with a total amount of 50 µL 0.2M HCl (Jenk et al., 2007). Afterwards the filters are left in a class 100 clean bench for 1h to allow

potentially present carbonates to be transformed into $CO_2$ by reaction with the HCl, followed by rinsing with 5 ml ultra-pure water to entirely remove remaining HCl. The filters are left again for 1h to reach complete dryness, packed in aluminium foil and kept frozen until analysis, for which filters are taken out of the freezer to let them reach ambient temperature (at least half an hour). Details regarding OC and EC separation, AMS [14]C analysis and improvements achieved since the first applications will be discussed in Sections 3 and 4.





## 3 Recent optimization in OC/EC separation and AMS analysis

In previous ice core dating applications using [14]C of WIOC (Jenk et al., 2009; Sigl et al., 2009; Kellerhals et al., 2010; Herren et al., 2013; Zapf et al., 2013), the OC and EC combustion was performed with the Two-step Heating system for the EC/OC Determination Of Radiocarbon in the Environment apparatus (THEODORE), developed for aerosol applications (Szidat et al., 2004b). The combustion was conducted in a stream of oxygen for the controlled separation of OC and EC fractions. The temperature for OC separation was set at 340°C, while for recovery of EC the temperature was then increased to 650°C. The $CO_2$ produced by oxidation during the combustion was cryogenically trapped, manometrically quantified and sealed in glass ampoules (Szidat et al., 2004b). In the earliest application described by Jenk et al. (2006) the $CO_2$ subsequently had to be transformed to filamentous carbon (graphitisation) using manganese granules and cobalt powder for final AMS [14]C analysis. This was initially performed at the ETH AMS facility (TANDY, 500 kV pelletron compact AMS system) (Synal et al., 2000). Since 2006, the 200kV compact AMS (MIni radioCArbon DAting System, MICADAS) has been operational at the ETH (Synal et al., 2007). The MICADAS is equipped with a gas ion source and a Gas Introduction interface System (GIS) (Ruff et al., 2007; Synal et al., 2007), allowing measurements of [14]C directly in $CO_2$ with an uncertainty level as low as 1% (Ruff et al., 2010). The GIS includes a gas-tight syringe for the $CO_2$ injection into the ion source (Ruff et al., 2010), with a maximum capacity of 1.3 ml of $CO_2$ as ~5% mixing ratios in helium (equivalent to 100 μg of carbon). The position of the syringe plunger is automatically adjusted according to the sample size as well as the helium flow carrying the sample to the ion source. With this, the tranformation of gaseous $CO_2$ to solid graphite targets became needless (Sigl et al., 2009). Instead, the glass ampoules sealed after the combustion of the filters with the THEODORE system were opened in a designated cracker, an integral part of the GIS (Ruff et al., 2007), and the resulting $CO_2$-He mixture could directly be fed into the MICADAS ion source.

The main advantages of switching from solid to gaseous targets were: 1. a decrease in the number of necessary preparation steps and the associated risk of lost samples from incomplete graphitisation, 2. a higher sample throughput, 3. a reduction in the variability and overall blank contribution as well as 4. the elimination of the correction applied to account for fractionation during the graphitisation step, which contributed with around 10% to the overall uncertainty (Jenk et al., 2007). As will be discussed in Section 4, a precision increase is one of the main challenges for improving the method.

Since spring 2013, [14]C analysis is performed with a MICADAS installed at the Laboratory for the Analysis of Radiocarbon with AMS (LARA laboratory) of the University of Bern, also equipped with a GIS interface (Szidat et al., 2014). There, an improvement was recently achieved by replacing the THEODORE with a commercial combustion system, which is a thermo-optical OC/EC analyzer (Model4L, Sunset Laboratory Inc., USA), normally used for aerosol OC/EC separation and source apportionment studies (Zhang et al., 2012; Zhang et al., 2013; Zhang et al., 2014; Zotter et al., 2014). Similar as in the THEODORE system, the carbonaceous particles are combusted in a stream of pure oxygen. The Sunset instrument is specially equipped with a non-dispersive infrared (NDIR) cell to quantify the $CO_2$ produced during the combustion. The combustion process in the Sunset system follows a well-established protocol (Swiss 4S) for the thermal separation of OC and EC fractions under controlled conditions (Zhang et al., 2012). To avoid potential damage of the infrared cell detector by residual HCl, the final rinsing of the filters after adding HCl for carbonates removal was introduced (see Section 2). Tests with blank filters and standard materials were performed to exclude any potential contamination from this additional step. The Sunset instrument is directly coupled to the zeolite trap of the GIS (Ruff et al., 2010), which allows online [14]C measurements of the carbonaceous fractions separated in the Sunset system (Agrios et al., 2015). When combusted, the gaseous carbonaceous species pass through a $MnO_2$ bed heated to 850°C for completing the oxidation to $CO_2$, which is further transported by helium to the zeolite trap. This trap is then heated up to 500°C to release the $CO_2$ to the gas-tight syringe for final injection into the AMS ion source (Ruff et al., 2007; Synal et al., 2007).

The newly coupled Sunset-GIS-AMS system has major advantages compared to the old setup. The OC/EC separation in the THEODORE was relatively time consuming and only four ice samples could be processed per day. Two more days were



needed to produce all the standards and blanks required for AMS calibration and for quality control and graphitisation (Jenk et al., 2007). Besides the disantvantages of solid graphite targets described before, there is also a risk of losing samples

during the delicate phase of flame-sealing the ampoules and later on when scratching them to allow a clean break in the automated GIS cracker. With the online coupling of the Sunset, this risk is completely removed. Further the preparation and measurement time is significantly reduced because there is no need for offline combustion resulting in a total measurement time of approximately 35 min per sample only. In addition, it not only allows for an automated protocol of standard injection for AMS calibration, but also offers the possibility for easy and regular (daily) survey of the $^{14}$C background in the entire

process line (Sunset-GIS-AMS) by analysis of variably sized standards and blanks if required (see last paragraph of this Section). Finally, the Sunset system enables continuous monitoring of the combustion process, reducing a potential bias due to charring, and the standardized and automated combustion protocol (Swiss 4S) ensures high reproducibility increasing the overall precision.

With the current setup, the $^{14}$C/$^{12}$C ratio of the samples is background substracted based on the AMS calibration using fossil

sodium acetate ($^{14}$C dead, NaOAc, p.a., Merck, Germany) and the reference material NIST standard oxalic acid II (modern, SRM 4990C) corrected for decay between 1950 and year of measurement, and normalized to account for mass fractionation in the AMS by $\delta^{13}$C simultaneously measured (Wacker et al., 2010). All results are expressed as Fraction Modern (F$^{14}$C), which is the deviation of the $^{14}$C/$^{12}$C ratio of the sample from that of the modern standard. $^{14}$C ages (BP) are calibrated using OxCal v4.2.4 (Bronk Ramsey and Lee, 2013) with the Northern (IntCal13) or Southern Hemisphere (ShCal13) calibration

curves (Hogg et al., 2013), depending on the sample site location. Calibrated dates are given in years before present (cal BP, with BP = 1950) with 1 σ uncertainty range (Stuiver and Polach, 1977; Mook and van der Plicht, 1999). For simplicity the ages dicussed in the text are given as the mean of this range ±1 σ. See Section 4 for further details regarding the applied corrections, $^{14}$C calibration and discussion of uncertainties.

To ensure comparability between previous data and the newly derived results, using the above described improved setup

configuration, $^{14}$C analysis was conducted on remaining pieces of samples, which were previously processed with the THEODORE setup. Two samples (Juv1 and Juv2) from the Juvfonne ice patch in Norway (Zapf et al., 2013) and two samples (Bel1 and Bel2) from an ice core drilled at Belukha glacier in the Siberian Altai (Aizen et al., 2016) were used, covering an age range from modern to more than 8000 cal BP. The OC masses were above 10 µg carbon, except for sample Juv2_Sunset with a carbon mass of 9 µg (Table 2), still resulting in more than 4500 $^{14}$C counts with a corresponding

uncertainty of the F$^{14}$C of 2%, which we consider sufficiently low for this comparison. At first, the obtained WIOC concentrations are discussed, which are assumed to agree as indicated by a carbon quantification test carried out on homogeneous aerosol filters using both combustion instruments (Zotter et al., 2014). As expected a good consistency was found for the WIOC concentrations in the Belukha ice core (Table 2), whereas a discrepancy was observed for the Juvfonne samples, probably related to the natural inhomogeneity of particles in this small-scale ice patch with a distinct ice

accumulation behaviour (see below). Concerning the $^{14}$C ages, a very good agreement is shown between all parallel samples (Figure 1). This is also true for the procedural blanks, both in term of carbon amount and F$^{14}$C. The THEODORE resulted in a blank contribution of 1.41 ± 0.69 µg with F$^{14}$C of 0.64 ± 0.12, and the Sunset in 1.21 ± 0.51 µg with a F$^{14}$C of 0.73± 0.13. We therefore conclude that dating results obtained with the old THEODORE combustion setup (Jenk et al., 2009; Sigl et al., 2009; Herren et al., 2013; Zapf et al., 2013) and with the improved coupled Sunset-GIS-AMS system are in good agreement.

## 205 4 Radiocarbon dating uncertainties

First of all, the signal-to-noise ratio of the AMS measurement is defined by counting statistics. Generally, the smaller the sample, the shorter the measurement time, the higher the uncertainty. For defining the contamination contribution of the overall instrument setup (constant contamination) and the memory effect between subsequent samples of very different $^{14}$C



content and carbon mass (cross contamination), a series with varying amounts of solid grains of fossil NaOAc and the

modern reference material oxalic acid II was measured for its $^{14}$C content. The constant contaminant mass was estimated as 0.4 ± 0.2 µg carbon with a F$^{14}$C of 0.8 ± 0.4 and for the cross contamination 0.5 ± 0.4% of the carbon of the previous sample was found to mix with the next injection (Agrios et al., 2015).

The total carbon amounts in ice cores are rather low, in the µg/kg-range. Because of that, each step of sample preparation implies a potential risk of contamination with either modern or fossil carbon. Thus a large contribution to the final overall

uncertainty on the age is induced by the procedural blank correction, especially for small size samples. It is therefore crucial that cutting, melting and filtrating the ice results in the lowest possible procedural blank with a stable F$^{14}$C value to ensure a high and stable signal-to-blank ratio for obtaining reliable results with the smallest possible uncertainties. Procedural blanks were estimated using artificial ice blocks of frozen ultra-pure water, treated in the same way as real ice samples (Jenk et al., 2007). Blanks were usually prepared together with samples and their analysis was performed during every AMS

measurement session. The average overall procedural blank is 1.34 ± 0.62 µg carbon with a F$^{14}$C of 0.69 ± 0.13 (100 and 54 measurements, respectively, performed over a 10-year period). These values are consistent with previously reported results (Jenk et al., 2007; Sigl et al., 2009), indicating the long-term stability of the procedural blanks.

In summary, all the corrections have the strongest effect on low carbon mass samples, resulting in the largest dating uncertainties. Further, such small samples can only be measured for a short period of time, with reduced stability of the $^{12}$C

current, additionally worsening of the signal-to-noise ratio. Low carbon mass samples of old age contain even a lower number of $^{14}$C compared to younger samples due to radioactive decay and are affected the most. Among all uncertainties described, the correction for the procedural blank contributes typically around 60%. As an example, for hypothetical samples with a WIOC mass of 5 or 10 µg, the resulting uncertainty for 1000 year old ice would be ± 600 yrs or ± 250 yrs and for 8000 year old ice ± 1600 yrs or ± 700 yrs, respectively. Hence by doubling the mass, the uncertainty is reduced by more than

50%. We therefore generally discuss dating results only for sample masses larger than ~10 µg WIOC, which have an acceptable age uncertainty in the range of 10-20%.

While calibrating the ages with the OxCal, a sequence constraint can be applied based on the assumption of a monotonous increase of age with depth (Bronk Ramsey, 2008). This approach often leads to a reduction of the final uncertainty, which however strongly depends on the sample resolution with depth, see example in Jenk et al. (2009).


## 5 Validation of the dating accuracy

### 5.1. First attempts

Validating the accuracy of the here described approach for $^{14}$C dating of ice is a challenging task since it requires ice samples with known ages, preferentially covering a large age range.

First attempts for validation by dating ice from Greenland with an age determined by annual layer counting failed, because WIOC concentrations are an order of magnitude lower compared to ice from glaciers located closer to biogenic emission sources (Figure 2). Large ice samples were thus needed, nevertheless resulting in small amounts of carbon. Our preparation method is not optimised for such sample sizes, and the required pooling of several pieces of ice may have induced a higher procedural blank. As a result $^{14}$C ages tended to be biased by the procedural blank value (Sigl et al., 2009). $^{14}$C ages of the

Fiescherhorn ice core (Jenk et al., 2006) ranged from modern values to 1000 years, thus reasonably matching the age of the ice older than AD 1800 obtained by annual layer counting. For the ice core from Mercedario (31.98° S, 70.13° W; 6100 m a.s.l.) the deepest core sections show ages of <550 and 320–1120 cal BP, respectively, well in line with a tentative chronology based on annual layer counting (Sigl et al., 2009). However, considering the relatively large uncertainty of our method if compared to conventional $^{14}$C dating typically derived from samples with much larger carbon masses and the

flatness of the $^{14}$C calibration curve between around 500 and 0 cal BP such samples of rather young ages are not ideal for a





precise validation. Two samples from the Illimani ice core, bracketing the AD 1258 volcanic eruption time marker resulted in a combined calibrated age of AD 1050±70 (1 σ) overestimating the expected age by 200±70 years (1 σ). This would be an acceptable accuracy if applicable to several thousand years old ice (Sigl et al. 2009).

Overall these were first indications that the [14]C method gives reliable ages. Meanwhile we have had access to independently
dated ice from the Juvfonne ice patch and the Quelccaya ice cap, dated a fly which we discovered in the Tsambagarav ice core, and dated ice cores from Mt. Ortles glacier, in which a larch leaf was found, altogether allowing a more robust validation as outlined in the following.

### 5.2. Recent validation

Juvfonne is a small perennial ice patch in the Jotunheimen Mountains in central southern Norway (61.68° N, 8.35° E). In
May 2010, a 30-m-long ice tunnel was excavated, revealing several up to 5 cm thick dark layers containing organic residues, which were interpreted as previous ice-patch surfaces and conventionally [14]C dated (Nesje et al., 2012). We received two samples of clear ice adjacent to the organic-rich layers and a surface sample (JUV 1, JUV 2, JUV 3, Table 3). The results derived using WIOC agreed well with the corresponding, conventionally dated [14]C ages with an age range between modern and 2900 cal BP (Zapf et al., 2013). In summer 2015 we collected additional clear ice samples adjacent to a 6600 years old
plant fragment layer found at the base of a new tunnel excavated in 2012 and extending deeper into the ice patch (Ødegård et al., 2016). Four ice blocks were collected and afterwards subdivided in two sub-samples each. Ice block 1 (JUV 0_1 and JUV 0_2) was taken adjacent to the plant fragment layer, ice block 2 (JUV 0_3 and JUV 0_4), ice block 3 (JUV 0_5 and JUV 0_6) and ice block 4 (JUV 0_7 and JUV 0_8) at the bottom of the wall, a few cm below the plant fragment layer. JUV 0_1 and JUV 0_2 yielded an average age of 7112 ± 143 cal BP, which is in good agreement with the age of the plant
fragment layer of 6608 ± 53 cal BP. The other six samples are significantly older (7595 ± 75 cal BP, Table 3), which is reasonable since they were collected below the plant fragment layer.

Three sections of the ice core from the Quelccaya Summit Dome drilled in 2003 (QSD, Peruvian Andes, 168.68 m, 13°56'S, 70°50'W, 5670 m a.s.l.) were kindly provided by Lonnie Thompson, Ohio State University. The entire ice core was dated by annual layer counting indicating an age of 1800 years at the bottom (Thompson et al., 2013). Intentionally we received the
samples without knowing their ages or depths in order to have the opportunity to perform a "blind test". The three sections were not decontaminated as usual, but only rinsed with ultra-pure water, because the amount was not large enough for removing the outer layer mechanically. As shown in Figure 3 (see also Table 4 for the results) the resulting calibrated ages agree very well with the ages based on annual layer counting (L. Thompson, personal communication 2015).

Recently a number of core segments of the previously dated Tsambagarav ice core (Herren et al., 2013) were resampled. In
segment 102 a tiny insect (Figure 4) was found and immediately separated from the ice matrix. Since it was small, a conventional [14]C analysis was not suitable and instead the Sunset-AMS system was deployed. The ice section containing the fly was melted, possible contamination from carbonates and humic acids were removed by an acid-base-acid treatment at 40°C (Szidat et al., 2014), the fly was dried, placed onto a quartz fibre filter and combusted in the Sunset, resulting in 13 µg of carbon. The age of 3442 ± 191 cal BP (BE-5013.1.1) is in perfect agreement with the age of WIOC from this ice segment
of 3495 ± 225 cal BP (Herren et al., 2013) (Figure 3).

Additionally, we dated three sections from a set of ice cores drilled in 2011 on Mt. Ortles (see Table 1 for location) for which a preliminary age of 2612 ± 101 cal BP was derived by conventional [14]C dating of a larch leaf found at 73.2 m depth (57.8 m weq, ~1.5 m above bedrock) (Gabrielli et al., 2016, this issue). Every section was horizontally divided in three sub-samples (top, middle, bottom). The ages obtained for the sub-samples of the sections at 68.61 m (53.82 m weq) depth (core
#1) and at 71.25 m (56.17 m weq) depth (core #3) were not significantly different from each other, respectively. Accordingly the derived ages were combined using the corresponding function in OxCal v4.2.4 ([14]C date combination). On the contrary the ages of the three sub-samples from the deepest section at 74.13 m (58.62 m weq) (core #3) significantly increased with





depth, implying strong glacier thinning close to bedrock (see also Gabrielli et al., 2016, this issue). Our WIOC [14]C ages obtained for the Mt. Ortles ice core agree well with the age of the larch leaf assuming an exponential increase of age with depth (Figure 5).

The scatter plot in Figure 3 summarizes the different validation experiments described above. The results for the Mt. Ortles ice core were not included because larch leaf and WIOC samples were extracted from depths of significantly different ages. As shown, within the uncertainties, the [14]C ages fall onto the 1:1 line in the age range from ~700-3500 cal BP, convincingly demonstrating good accuracy of our method.

## 6 Applications and current potential of the [14]C method for dating glacier ice

Over the last 10 years the deepest parts of several ice cores have been dated applying the presented WIOC [14]C method. To illustrate the current potential of the method with respect to the time period accessible we compiled five ice core chronologies in Figure 6. The sites differ in recent net annual snow accumulation and ice thickness (in brackets): Tsambagarav ice cap in the Mongolian Altai 0.33 m weq (72 m) (Herren et al., 2013), Belukha glacier in the Siberian Altai 0.34 m weq (172 m) (Aizen et al., 2016), Colle Gnifetti glacier in the European Alps 0.46 m weq (80 m) (Jenk et al., 2009), Illimani glacier in the Bolivian Andes 0.58 m weq (138.7 m) (Kellerhals et al., 2010), Mt. Ortles glacier 0.85 m weq (75 m) (Gabrielli et al., 2016, this issue). All of these are cold glaciers and frozen to the bedrock with the exception of Mt. Ortles glacier, which is polythermal and basal sliding can not totally be excluded at least for certain periods of time (Gabrielli et al., 2016, this issue). To derive a continuous age depth relationship, a two parameter flow model (Nye, 1963; Bolzan, 1985; Thompson et al., 1990) was applied for Colle Gniffetti (Jenk et al., 2009), Illimani (Kellerhals et at., 2010) and here also for the core from Belukka using the data presented in Aizen et al. (2016). A different approach as discussed below, was implemented for the ice cores from the Tsambagarav ice cap (Herren et al., 2013) and the glacier on Mt. Ortles (see also Gabrielli et al., 2016, this issue). The two parameter model is based on a simple analytical expression for the decrease of the annual layer thickness $L_{(z)}$ (m weq) with depth:

$$L_{(z)} = b(1 - \frac{z}{H})^{p+1}$$

where z is depth (m weq), H the glacier thickness (m weq), b the annual accumulation (m weq) and p a thinning parameter (dimensionless). The age T(z) as a function of depth can be calculated when the inverse layer thickness is integrated over depth:

$$T_{(z)} = \int \frac{dz}{L_{(z)}} = \frac{1}{b} \int (1 - \frac{z}{H})^{-p-1} dz$$

Solving the integral and setting the age at the surface to be T(0) = 0, the final age-depth relation is obtained:

$$T_{(z)} = \frac{H}{bp} [(1 - \frac{z}{H})^{-p} - 1]$$

The thinning rate (vertical strain rate) is the first derivative of the layer thickness:

$$L'_{(z)} = \frac{dL_{(z)}}{dz} = -\frac{b(p + 1)}{H} (1 - \frac{z}{H})^{p}$$

The model has two degrees of freedom, the net accumulation rate and the thinning parameter p both assumed to be constant over time. This allows to fit the model by a least squares approach through selected reference horizons if the glacier thickness H is known (if drilled to bedrock) or can be reasonably well estimated (e.g. from radar sounding). In order to not overweight the data from the deepest horizons, the model is fitted using the logarithms of the age values. For the ice cores from Colle Gnifetti (Jenk et al., 2009), Illimani (Kellerhals et al., 2010) and Belukha (Aizen et al., 2016) these ages were based on annual layer counting, identification of reference horizons (radioactive fallout and well-known volcanic eruptions)





and [14]C dates. The data is summarized in Table 1. In Figure 6, only reference horizons and [14]C dates were included for simplification.

In summary, a reasonable fit was achieved for these three glaciers and the derived annual net accumulations (Colle Gnifetti 0.45±0.03 m weq, Belukha 0.36 ± 0.03 m weq, Illimani 0.58 m weq) are comparable with the values previously published

(see above) which were determined by surface measurements or with the estimated accumulation using the upper (not [14]C dated) age horizons when accounted for layer thinning by applying the model described by Nye (1963).

However, with the two parameter model no fit could be achieved for the ice cores from Tsambagarav and Mt. Ortles. Whereas Tsambagarav also is a cold glacier, Mt. Ortles is polythermal. For Tsambagarav, a fit is only possible if additional degree of freedom is given to account for variations in the net accumulation rate. Opposite to that, a reasonable fit for the Mt.

Ortles ice core can only be obtained if the thinning parameter p is allowed to increase with depth. In both cases, a purely empirical approach of fitting the age horizons was chosen to yield the age-depth relationship (due to the lack of absolute time markers prior to 1958, [210]Pb dated horizons with a larger uncertainty compared to the age of time markers were used for Mt. Ortles). For Tsambagarav a combination of different polynomial functions was used (Herren et al., 2013), whereas a slightly more sophisticated approach by means of Monte Carlo simulation was applied for Mt. Ortles allowing an objective

uncertainty estimate for each depth defined by the density of dating horizons and their individual uncertainty (Gabrielli et al., 2016, this issue). Such a purely empirical approach is justified given the high confidence assigned to the determined ages for the dated horizons. For Tsambagarav, the strong variation in net accumulation was consistent with precipitation changes derived from lake sediment studies in the Altai (Herren et al., 2013). Mt. Ortles glacier is polythermal with temperate conditions in the upper part and still relatively warm ice with -2.8°C near bedrock. We hypothesize that the faster horizontal

velocity of the warm ice causes exceptional horizontal stress (internal horizontal deformation) on the ice frozen to the bedrock, resulting in stronger thinning.

As shown in Figure 6, the time period dated with [14]C ranges from 200 to more than 10000 yrs. Due to their uncertainty, [14]C ages derived by our method cannot compete with the conventional methods for dating ice that is only a few centuries old. The strength of [14]C dating using WIOC is that it allows obtaining absolute ages from principally every piece of ice. This is

especially valuable for glaciers not containing the last glacial/interglacial transition, as Tsambagarav and Mt. Ortles (Gabrielli et al., 2016, this issue), since in such cases not even wiggle matching of the transition signal with other dated archives is possible. Anyhow, an absolute dating method is superior to wiggle matching, which is not necessarily reliable. For example, a depletion in $\delta^{18}$O presumably indicating the LGM-Holocene transition might not always be a true atmospheric signal, but can be caused by unknown mechanisms potentially happening close to bedrock (Jenk et al., 2009).

All five examples show strong thinning towards bedrock and oldest ages obtained were in the range of 10000 years. In these cases the age limit was determined by the depth resolution, since some hundred grams of ice are required and not by the half-life of [14]C of 5730 yrs. With this strong thinning the [14]C age of the deepest sample may represent a mixed age of ice having a large age distribution.

Since an absolute WIOC mass of ~10 µg is needed to achieve a [14]C dating with reasonably low uncertainty, the overall

applicability of the method essentially depends on the WIOC concentration in the ice and the ice mass available. Figure 2 summarizes WIOC concentrations determined in ice from various locations around the globe. In general, mid-latitude and low-latitude glaciers contain sufficient WIOC from 21 to 295 µg/kg, allowing dating with less than 1 kg of ice. The highest concentration was found at Juvfonne ice patch which is small and located a low elevation and therefore by far closest to biogenic emission sources. WIOC concentrations might be further elevated due to meltwater and superimposed ice

formation, enriching water-insoluble particles in the surface layer present at that time. Lowest concentrations of only 2 to 15 µg/kg WIOC were observed in polar snow and ice from Greenland and Antarctica. For this concentration range a reliable dating is impossible with the current method capability.

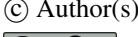



**7 Conclusions**

Since the introduction about 10 years ago of the [14]C dating technique for glacier ice, utilizing the WIOC fraction of
carbonaceous aerosol particles embedded in the ice matrix, major improvements in separating the OC from the EC fraction
and in AMS technology have been achieved. The new configuration with direct coupling of a commercial thermo-optical
OC/EC analyser to the gas ion source of the MICADAS AMS via its gas introduction interface has two major advantages.
First, the measurement time was significantly reduced to approximately 35 min per sample. Second, the implemented
automated protocol allows for a controlled routine analysis with high reproducibility and a stable blank, thereby increasing
the overall precision.

The presented [14]C WIOC dating method was validated by determining the age of independently dated ice samples. It
principally allows absolute and accurate dating of any piece of ice containing sufficient WIOC. With the current set-up, the
age of samples with a minimum of ~10 µg WIOC can be determined with satisfying precision of about 10 to 20%, depending
on the age. This requires about 100 to 500 g of ice considering the WIOC concentrations typically found in mid- and low
latitude glaciers. Dating polar ice with satisfactory age uncertainties is still not possible since WIOC concentrations are
around one order of magnitude lower. This would require further reduction of the procedural blank for such samples
requiring larger ice volumes which potentially could be achieved by an additional, specifically designed sample preparation
setup for such kind of samples.

The [14]C method is suitable for dating ice with ages from 200 to more than 10000 yrs. Whereas for a few century old ice the
conventional dating methods are typically higher in precision, the [14]C WIOC method presents the only option for obtaining
reliable continuous time scales for the older and deeper ice core sections of mountain glaciers. This is not only crucial for
interpreting the embedded environmental and climatic history, but gives additional insight into glacier flow dynamics close
to bedrock as demonstrated by the depth-age scales derived from [14]C dating of ice cores from various mid- and low latitude
glaciers. Also, it can reveal information about the time of glacier formation.

**Acknowledgements**

This work was supported by the Swiss National Science Foundation (200020_144388) and by the Oeschger Centre for
Climate Change Research of the University of Bern.

**Author contribution**

Manuscript written by C.U., T.M.J. and M.S. with editing by S.Z.. Sample preparation and [14]C measurements performed by
A.Z. and C.U. with expert supervision of G.S. and T.M.J..





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

Methods in Physics Research Section B: Beam Interactions with Materials and Atoms, 259, No. 1, 7-13,
10.1016/j.nimb.2007.01.138, 2007.

Szidat, S., Jenk, T. M., Gäggeler, H. W., Synal, H. A., Fisseha, R., Baltensperger, U., Kalberer, M., Samburova, V., Wacker,
L., Saurer, M., Schwikowski, M., and Hajdas, I.: Source apportionment of aerosols by $^{14}$C measurements in different
carbonaceous particle fractions, Radiocarbon, 46, No. 1, 2004a.

Szidat, S., Jenk, T. M., Gäggeler, H. W., Synal, H. A., Hajdas, I., Bonani, G., and Saurer, M.: THEODORE, a two-step
heating system for the EC/OC determination of radiocarbon ($^{14}$C) in the environment, Nuclear Instruments and Methods in
Physics Research Section B: Beam Interactions with Materials and Atoms, 223-224, 829-836, 10.1016/j.nimb.2004.04.153,
2004b.

Szidat, S., Salazar, G. A., Battaglia, M., Wacker, L., Synal, H.-A., Vogel, E., and Türler, A.: $^{14}$C analyses and sample
preparation at the new Bern Laboratory for the Analyses of Radiocarbon with AMS (LARA), Radiocarbon, 56, No. 2, 561–
566, 2014.

Thompson, L. G., Davis, M. E., Mosley-Thompson, E., Sowers, T. A., Henderson, K. A., Zagorodnov, V. S., Lin, P. N.,
Mikhalenko, V. N., Campen, R. K., Bolzan, J. F., Cole-Dai, J., and Francou, B.: A 25,000-Year Tropical Climate History
from Bolivian Ice Cores, Science, 282, No. 5395, 1858-1864, 10.1126/science.282.5395.1858, 1998.

Thompson, L. G., Mosley-Thompson, E., Davis, M., Bolzan, J. F., Dai, J., Klein, L., Gundestrup, N., Yao, T., Wu, X., and
Xie, Z.: Glacial stage ice-core records from the subtropical Dunde ice cap, China, Annals of Glaciology, 14, 288-297, 1990.

Thompson, L. G., Mosley-Thompson, E., Davis, M., Henderson, K., Brecher, H. H., Zagorodnov, V. S., Mashiotta, T. A.,
Lin, P. N., Mikhalenko, V. N., Hardy, D. R., and Beer, J.: Kilimanjaro ice core records: Evidence of Holocene climate
change in Tropical Africa, Science, 298, No. 5593, 589-593, 2002.

Thompson, L. G., Mosley-Thompson, E., Davis, M. E., Zagorodnov, V. S., Howat, I. M., Mikhalenko, V. N., and Lin, P. N.:
Annually Resolved Ice Core Records of Tropical Climate Variability over the Past ~1800 Years, Science Express, 2013.

Wacker, L., Bonani, G., Friedrich, M., Hajdas, I., Kromer, B., Němec, M., Ruff, M., Suter, M., Synal, H. A., and
Vockenhuber, C.: MICADAS: routine and high-precision radiocarbon dating, Radiocarbon, 52, No. 2-3, 252-262, 2010.

Zapf, A., Nesje, A., Szidat, S., Wacker, L., and Schwikowski, M.: $^{14}$C measurements of ice samples from the Juvfonne ice
tunnel, Jotunheimen, Southern Norway—validation of a $^{14}$C dating technique for glacier ice, Radiocarbon, 55, No. 2–3, 571–
578, 2013.

Zhang, Y.-L., Li, J., Zhang, G., Zotter, P., Huang, R.-J., Tang, J.-H., Wacker, L., Prévôt, A. S. H., and Szidat, S.:
Radiocarbon-Based Source Apportionment of Carbonaceous Aerosols at a Regional Background Site on Hainan Island,
South China, Environmental Science & Technology, 48, 2651–2659, 2014.

Zhang, Y.-L., Perron, N., Ciobanu, V. G., Zotter, P., Minguillón, M. C., Wacker, L., Prévôt, A. S. H., Baltensperger, U., and
Szidat, S.: On the isolation of OC and EC and the optimal strategy of radiocarbon-based source apportionment of
carbonaceous aerosols, Atmos Chem Phys, 12, No. 22, 10841-10856, 10.5194/acp-12-10841-2012, 2012.





Zhang, Y.-L., Zotter, P., Perron, N., Prévôt, A. S. H., Wacker, L., and Szidat, S.: Fossil and non-fossil sources of different
       carbonaceous fractions in fine and coarse particles by radiocarbon measurement, Radiocarbon, 55, No. 3-4, 2013.

       Zotter, P., Ciobanu, V. G., Zhang, Y. L., El-Haddad, I., Macchia, M., Daellenbach, R., Salazar, G. A., Huang, R.-J., Wacker,
       L., Hueglin, C., Piazzalunga, A., Fermo, P., Schwikowski, M., Baltensperger, U., Szidat, S., and Prévôt, A. S. H.:
       Radiocarbon analysis of elemental and organic carbon in Switzerland during winter-smog episodes from 2008 to 2012 – Part
1: Source apportionment and spatial variability, Atmos Chem Phys, 14, 13551–13570, 2014.



**Table 1:** Characteristics of the sites discussed and the respective dating approach. ALC stands for Annual Layer Counting, RH for Reference Horizons and $^{210}$Pb, $^{3}$H, and $^{14}$C for nuclear dating. 2p model (two parameter model), MC (Monte Carlo simulation) and EF (exponential fit) denotes the applied approach to finally derive a continuous age-depth relationship (see Section 6 for details).

| Site | Coordinates Elevation | Location | Dating approach | Time span (years) | References |
|---|---|---|---|---|---|
| Belukha | 49.80°N, 86.55°E 4115 m a.s.l. | Altai Mountains, Russia | ALC, RH, $^{3}$H, $^{14}$C, 2p model | ~9100 | Aizen et al., 2016 |
| Colle Gnifetti | 45.93°N, 7.88°E 4450 m a.s.l. | Western Alps, Swiss-Italian border | ALC, RH, $^{3}$H, $^{210}$Pb, $^{14}$C, 2p model | >15200 | Jenk et al., 2009 |
| Juvfonne | 61.68˚N, 8.35E 1916 m a.s.l. | Jotunheimen Mountains, Norway | $^{14}$C of plant fragment and WIOC | ~7600 | Zapf et al 2013 Ødegård et al., 2016 |
| Illimani | 17.03°S, 68.28°W 6300 m a.s.l. | Andes, Bolivia | ALC, RH, $^{3}$H, $^{210}$Pb, $^{14}$C, 2p model | ~12700 | Sigl et al., 2009 Kellerhals et al., 2010 |
| Mt. Ortles | 46.51°N, 10.54°E 3905 m a.s.l. | Eastern Alps, Italy | ALC, RH, $^{3}$H, $^{210}$Pb, $^{14}$C, MC | ~6900 | Gabrielli et al., 2016 |
| Quelccaya | 13.93°S, 70.83°W 5670 m a.s.l. | Andes, Peru | ALC, $^{14}$C | ~1800 | Thompson et al., 2013 |
| Tsambagarav | 48.66°N, 90.86°E 4130 m a.s.l. | Altai Mountains, Mongolia | ALC, RH, $^{3}$H, $^{210}$Pb, $^{14}$C, EF | ~6100 | Herren et al., 2013 |

**Table 2:** Samples analysed for the comparararability test for OC/EC separation using the THEODORE apparatus and the Sunset OC/EC analyzer directly coupled to the AMS, with WIOC masses and concentrations. Calibrated ages (cal BP) denotes the 1 σ range.

| Sample ID | AMS Lab. No. | WIOC mass (µg) | WIOC concentration µg/kg ice | F$^{14}$C | $^{14}$C age (BP) | cal age (cal BP) |
|---|---|---|---|---|---|---|
| Juv1_THEODORE | ETH 42845.1.1 ETH 42847.1.1 ETH 42849.1.1 ETH 43446.1.1 | 44.4 | 176 | 1.134 ± 0.015 | -996 ± 52 | (-42 - -45) |
| Juv1_Sunset | BE 3683.1.1 BE 3701.1.1 | 45.5 | 119 | 1.159 ± 0.019 | -1175 ± 90 | (-42 - -8) |
| Juv2_THEODORE | ETH 43555.1.1 ETH 43557.1.1 | 17.6 | 60 | 0.742 ± 0.026 | 2227 ± 277 | (1904–2697) |
| Juv2_Sunset | BE 3679.1.1 | 9 | 33 | 0.751 ± 0.021 | 2300 ± 225 | (2065 - 2700) |
| Bel1_THEODORE | ETH 42841.1.1 | 18.3 | 63 | 0.771 ± 0.016 | 2090 ± 172 | (1900 - 2300) |
| Bel1_Sunset | BE 4282.1.1 | 15.4 | 61 | 0.743 ± 0.022 | 2430 ± 230 | ( 2160 - 2760) |
| Bel2_THEODORE | ETH 43448.1.1 | 14.9 | 47 | 0.425 ± 0.012 | 7329 ± 445 | ( 7680 - 8597) |
| Bel2_Sunset | BE 4175.1.1 | 17.8 | 48 | 0.388 ± 0.022 | 7611 ± 446 | ( 8009 - 8997) |





**Table 3:** Juvfonne samples analysed for method validation. JUV 1, JUV 2 and JUV 3 from the 2010 tunnel (Zapf et al., 2013; Ødegård et al., 2016) and JUV 0 from the 2012 tunnel (Ødegård et al., 2016).

565

| Sample ID | AMS Lab. No. | WIOC (µg) | Ice mass (g) | F$^{14}$C | $^{14}$C age (BP) | cal age (cal BP) |
|---|---|---|---|---|---|---|
| JUV 3_1 | ETH 42845.1.1 | 54.8 | 291.5 | 1.12±0.01 | -940±91 | |
| JUV 3_2 | ETH 42847.1.1 | 43.1 | 267.7 | 1.09±0.01 | -720±110 | |
| JUV 3_3 | ETH 42849.1.1 | 46.8 | 325.0 | 1.15±0.01 | -1160±100 | |
| JUV 3_4 | ETH 43446.1.1 | 43.4 | 207.7 | 1.16±0.02 | -1220±120 | |
| **JUV 3 (surface 2010)** | | | | **1.13±0.01** | **-996±52** | **modern** |
| JUV 2_1 | ETH 43443.1.1 | 27.3 | 215.1 | 0.88±0.02 | 1020±210 | |
| JUV 2_2 | ETH 43445.1.1 | 9.0 | 170.8 | 0.79±0.07 | 1870±670 | |
| JUV 2_3 | ETH 43559.1.1 | 16.5 | 257.4 | 0.87±0.04 | 1120±320 | |
| JUV 2_4 | ETH 45109.1.1 | 19.4 | 219.0 | 0.87±0.03 | 1130±280 | |
| **JUV 2 (2010)** | | | | **0.87±0.02** | **1116±146** | **(918-1237)** |
| **Organic remains, Poz-37879** | | | | **0.838±0.003** | **1420±30** | **(1300-1338)** |
| JUV 1_3 | ETH 43555.1.1 | 20.2 | 280.6 | 0.77±0.03 | 2144±300 | |
| JUV 1_4 | ETH 43557.1.1 | 9.2 | 214.0 | 0.72±0.06 | 2650±710 | |
| **JUV 1 (2010)** | | | | **0.76±0.03** | **2227±277** | **(1904-2697)** |
| | Poz-37878 | | | 0.826±0.003 | 1535±30 | |
| | Poz-36460 | | | 0.692±0.003 | 2960±30 | |
| **Organic remains mean** | | | | **0.76±0.04** | **2215±410** | **(1810-2750)** |
| JUV 0_1 | BE 4184.1.1 | 393.2 | 283.1 | 0.48±0.01 | 5905±248 | |
| JUV 0_2 | BE 4380.1.1 | 245.9 | 298.1 | 0.46±0.01 | 6293±137 | |
| **JUV 0-A (2015)** | | | | **0.47±0.01** | **6207±120** | **(6969-7255)** |
| **Organic remains, Poz-56955** | | | | **0.485±0.002** | **5800±40** | **(6556-6661)** |
| JUV 0_3 | BE 4185.1.1 | 219.4 | 207.9 | 0.44±0.01 | 6512±216 | |
| JUV 0_4 | BE 4381.1.1 | 182.4 | 188.3 | 0.44±0.01 | 6555±133 | |
| JUV 0_5 | BE 4186.1.1 | 238.3 | 226.9 | 0.40±0.01 | 7296±231 | |
| JUV 0_6 | BE 4382.1.1 | 36.4 | 184.2 | 0.44±0.01 | 6626±196 | |
| JUV 0_7 | BE 4187.1.1 | 262.2 | 200.4 | 0.40±0.01 | 7285± 18 | |
| JUV 0_8 | BE 4383.1.1 | 202.9 | 214.2 | 0.45±0.01 | 6396±229 | |
| **JUV 0-B (2015)** | | | | **0.43±0.01** | **6741± 9** | **(7519-7670)** |

**Table 4:** Quelccaya samples analysed for method validation. Calibrated ages (cal BP) denote the 1 σ range. ALC stands for
570   Annual Layer Counting.

| Sample | Depth (m) | AMS Lab. No. | WIOC (µg) | F$^{14}$C | $^{14}$C age (BP) | cal age (cal BP) | ALC (yrs BP) |
|---|---|---|---|---|---|---|---|
| 139-140 | 144.69-146.79 | BE 4336.1.1 | 15.1 | 0.89±0.03 | 954±237 | (575-970) | 730-788 |
| 149-150 | 155.21-157.31 | BE 4335.1.1 | 23.5 | 0.86±0.02 | 1220±171 | (979-1272) | 1072-1157 |
| 157-158 | 163.88-166.09 | BE 4337.1.1 | 14.4 | 0.80±0.02 | 1761±246 | (1378-1905) | 1439-1543 |



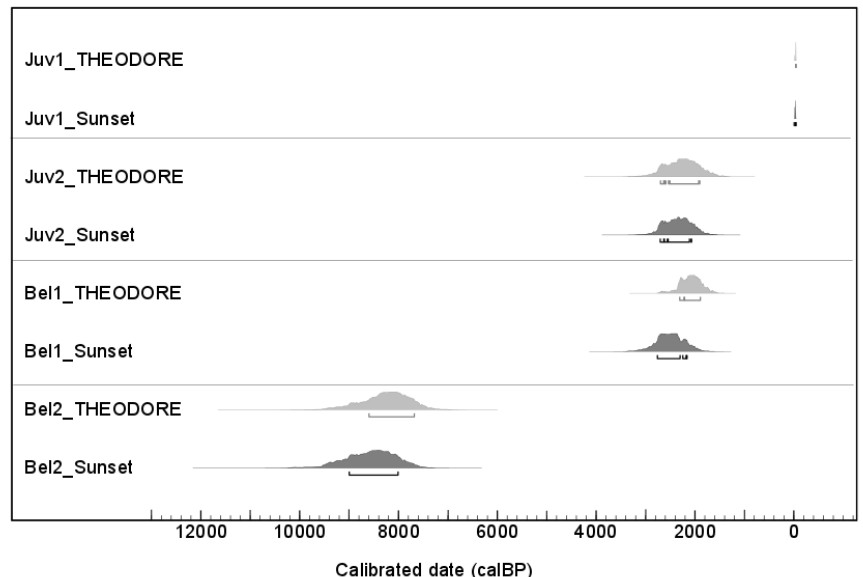

**Figure 1:** OxCal output for the comparararability test for OC/EC separation using the THEODORE apparatus and the Sunset
OC/EC analyzer directly coupled to the AMS. Bars below the age distributions indicate the 1 σ range. See Table 2 for
sample details.

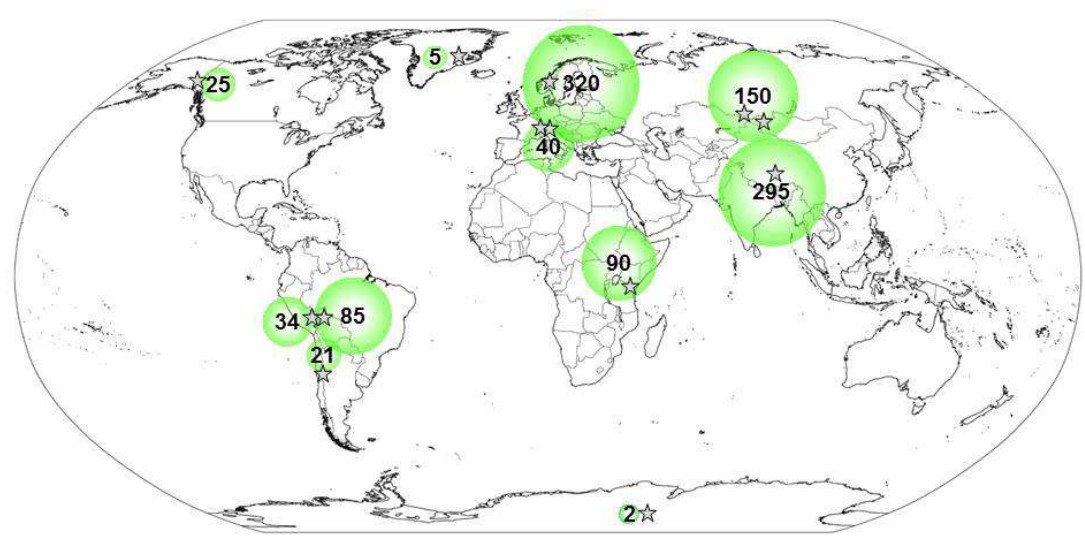

**Figure 2:** World map showing the sites from which ice samples were analysed with the $^{14}$C method (gray stars): Edziza,
Canada, 57.71° N 130.63° W; GRIP, Greenland, 72.59° N, 37.65° W, 3230 m a.s.l.; Juvfonne, Norway, 61.68 N, 8.35° E;
Colle Gnifetti, Switzerland, 45.93° N, 7.87° E; Mt. Ortles, Italy, 46.51° N, 10.54° E; Belukha, Russia, 49.80° N, 86.55° E;
Tsambagarav, Mongolia, 48.66° N, 90.86° E; Naimonanji, China 30.45° N, 81.54° E; Kilimanjaro, Tanzania, 3.06° S 37.34°
E; Quelccaya, Peru, 13.93° S, 70.83° W; Nevado Illimani, Bolivia, 16.03° S, 67.28° W; Mercedario, Argentina, 31.97° S,
70.12° W; Scharffenbergbotnen, Antartica, 74.00° S, 11.00° W. The average WIOC concentration in μg/kg at each site is
indicated with green bubbles.




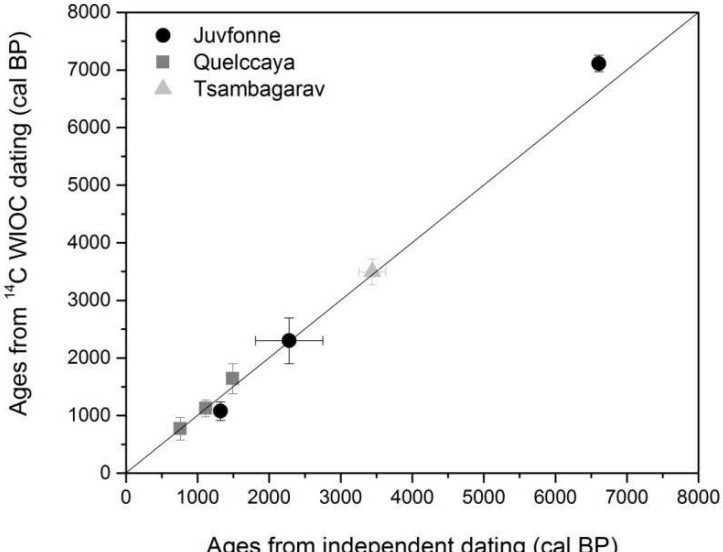

**Figure 3:** Scatter plot showing the ages obtained with the $^{14}$C method for independently dated ice, including the conventionally $^{14}$C dated Juvfonne plant fragment layers, (Ødegård et al., 2016), the $^{14}$C dated fly found in the Tsambagarav ice core, and the Quelccaya ice dated by annual layer counting, (Thompson et al., 2013). Error bars denote the 1 σ uncertainty.

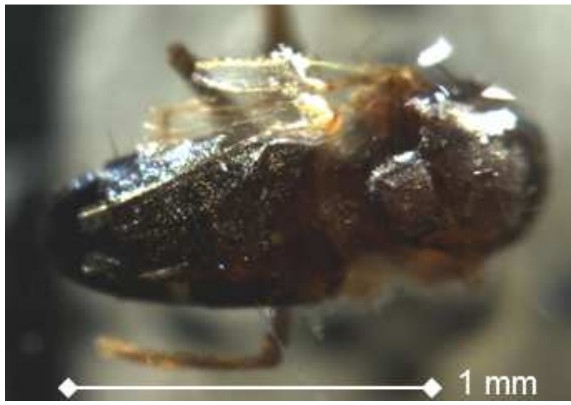

**Figure 4:** Photo of the fly found in segment 102 of the Tsambagarav ice core. The age of the fly was 3442±191 cal BP, while the surrounding ice yielded an age of 3495±225 cal BP.




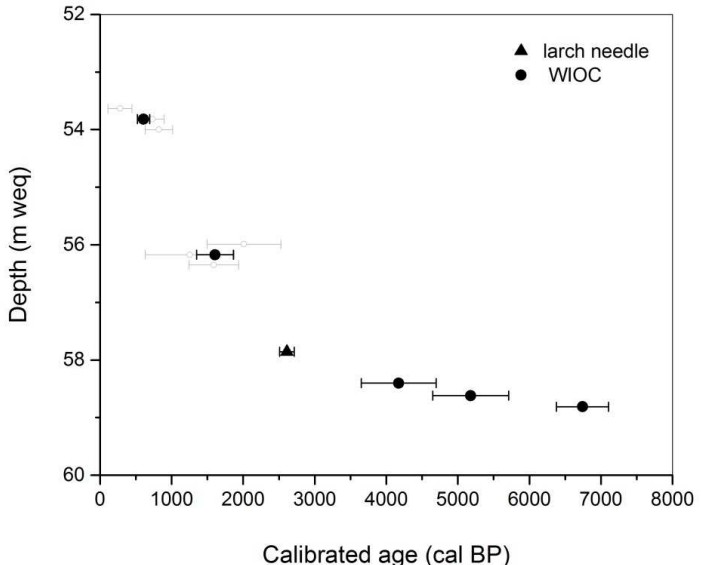

**Figure 5:** Dating of the bottom part of the Ortles ice core. Circles indicate the ages derived with the [14]C WIOC method and the triangle shows the age of the conventionally [14]C dated larch leaf found in the ice core (Gabrielli et al., 2016). Light grey circles show the ages obtained for the subsamples. Errors bars represent the 1 σ uncertainty.

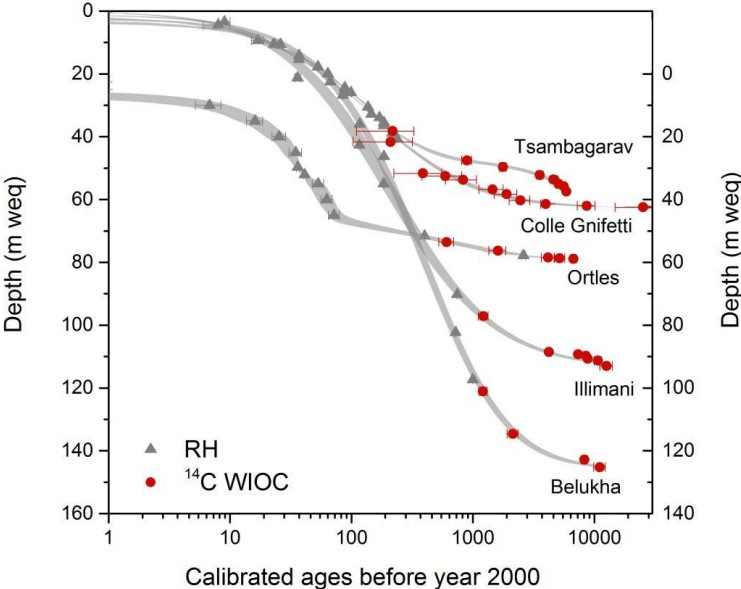

**Figure 6:** Compilation of age-depth relationships for five different ice cores. For simplicity only reference horizons and [14]C dates were included. Gray triangles indicate reference horizons (RH) and red circles the [14]C WIOC ages with uncertainties. For the Mt. Ortles core [210]Pb dated horizons with a larger uncertainty were used as RH due to the lack of absolute time markers prior to 1958; the gray triangle at 57.8 m weq depth is the conventional [14]C age of the larch leaf. Gray shaded areas represent the respective fit for retrieving a continuous age depth relationship. For sample details and the approach applied for fitting, see text and Table 1 (references therein). Note that for better visibility the curve for the Mt. Ortles glacier was shifted down by 20 m (right-hand y-axis).