# Peer review of "Radiocarbon dating of glacier ice: overview, optimisations, validation and potential"

_The Cryosphere, 2016_

## Referee Comment (RC1) · Anonymous Referee #1 · 23 Aug 2016

The authors present an overview of existing radiocarbon dating results from a suite of global ice cores, and new information on updated analysis techniques and validation exercises. On the whole, these approaches are invaluable for providing chronological information from deep portions of alpine cores, where other techniques fail yet there is still sufficient 14C available to provide reliable dates. I have no comments or concerns regarding the updated techniques or validation procedures demonstrated in the paper - all show the care and precision typical of this group. My main comment for the the authors to consider during revision is the overall presentation (and title) of the paper itself. As suggested by the title, and in the introduction (lines 90-106), on one hand the authors see this as a overview paper. But a second goal (lines 105-106) is to present recent optimizations in the analytical technique. These are not mutually exclusive, however it seems to me that the original portion of this paper is the validation exercise (section 5.2). Make no mistake, I find the optimization and validation to be novel, useful, and certainly worthy of publication here. I'm simply suggesting that the authors

refine the title and introduction to reflect this. The overview portions can stay as is - the summary in section 6 and figure 6 are quite useful for projecting how the technique could be useful in other situations.

---

## Referee Comment (RC2) · Anonymous Referee #2 · 7 Sep 2016

Radiocarbon dating of glacier ice is an important asset to ice core sciences in mid-latitudes when classical methods to derive age-depth models (layer counting, ice flow modelling, tephra-chronology) fail. However, suitable material for radiocarbon dating of macrofossils is sparse in the ice making it desirable to date other organic carbon compounds. Water-insoluble organic carbon (WIOC) has been shown to be a suitable candidate for radiocarbon dating of mid-latitude ice. Uglietti et al. review the efforts to develop the analytical methodology for radiocarbon dating of WIOC in ice, and test its accuracy. The group of Bern/PSI has been instrumental in developing the method so it seems only natural that they provide a review thereof and is certainly a valuable contribution to the literature of this topic. The paper is generally well written, thorough, suitable for the Cryosphere, and should be published. However, I have some comments that the authors might want to consider for the sake of clarity in the paper and some details the authors may want to check before publication.

[Figure]

1. My main comment is that in the context of analytical precision, I find it slightly misleading to discuss calibrated ages instead of 14C ages or fraction modern. The calibration of radiocarbon dates is a second methodology, introducing additional uncertainty due to the uncertainty of the calibration curve itself. Hence, 2 significantly different radiocarbon ages may lead to statistically indistinguishable calibrated ages. However, if the radiocarbon dates and their uncertainties are reliable, then 2 different radiocarbon ages are indicative of different calendar ages. This applies for example to table 2 and 3 and figure 1. These experiments refer to analytical accuracy of radiocarbon preparation and measurement, independent of the calibrated age and it is hence worth discussing (and showing) the differences in 14C years or F14C instead of calibrated years. A comparison of the 14C measurement results could for example be indicative whether the pure analytical precision does reflect the true uncertainty of the method. I would assume that the true uncertainty is somewhat larger, due to the inhomogeneous distribution of WIOC over an ice sample. This can for example be seen in Table 3 where the samples JUV 0_5/6 as well as JUV 0_7/8 yield significantly different radiocarbon ages, despite being from the same ice block. This questions whether these samples can be summarized to an error weighted mean and standard error as done in table 3. The final uncertainty of $\pm9$ 14C yrs for JUV 0_3-8 seems very small given the scatter of the individual measurements. A reduced Chi2 statistic of the sample pools in table 3 could be used to assess and discuss whether the uncertainties of single measurements are realistic. In a second step, it can then be discussed whether these uncertainties matter in terms of the absolute chronology, given that the calibration adds additional uncertainty.

2. Please check the data in the tables. I was confused seeing that the samples Bel2_THEODORE and Bel2_Sunset yield significantly different F14C values while their 14C ages agree. Using an F14C of 0.425 for Bel2_THEODORE I obtain a radiocarbon age of 6874 14C BP, as compared to 7329 14C BP given in table 2. So unless I missed something either the F14C or the 14C age of this sample is erroneous which might also impact on the calibrated ages shown in figure 1. Please check.

3. Throughout the manuscript the term "conventional" 14C dating is used to describe the dating of macrofossils. However, in the radiocarbon literature "conventional" 14C dating refers to 14C measurements using liquid scintillation and gas proportional counting techniques as opposed to AMS measurements. Please either use a different term than "conventional" or add a sentence defining how it is used in this paper.

L 45: please replace "nuclear" with "radiometric"

L 179-182: This is a very long sentence and a little unclear. Maybe divide it up into 2 sentences. Are the sample background corrected using OxII? I suppose the standard is used for normalization and not background correction? Please rephrase.

L 183: Please insert "relative" before deviation, as the samples are normalized to the standard.

L 183: "BP" is not explained at this point yet, but only in line 185. Please explain it here instead.

L 227-228: Are the uncertainties given in 14C years here? If so, please write "14C yrs" instead of years to be clear.

L 249 and following: See comment number 3. Please either define what you mean by conventional or use a different term throughout the manuscript instead.

L 252: This may be nitpicking, but AD 1258 – AD 1050 = 208 years, not 200 years.

L 259-271: Several times it is stated that the WIOC dates "agree well" with the macrofossil dates, while the 14C ages are indeed significantly different. I am not arguing against the general agreement but it would be great if you could add 1-2 sentences to make this more precise. Are the differences due to sampling differences (i.e., different ice layers have been sampled for macrofossils and WIOC)? If so, are the results in stratigraphic order?

L 351: Please write "climate wiggle matching" instead of just "wiggle matching" which

could also refer to the wiggle matching of radiocarbon dates.

L 357: Please add a reference to [Godwin, H. 1962. Nature, 195 (4845)] for the half-life of radiocarbon.

---

## Referee Comment (RC3) · Anonymous Referee #3 · 30 Sep 2016

The manuscript "Radiocarbon dating of glacier ice" by Uglietti et al. aims to give an overview of the actual 14C dating method applied by the authors group (i.e. at the Paul-Scherrer-Institute (PSI) for absolute dating of glacier ice via 14C analysis of the water insoluble organic carbon fraction WIOC. After the presentation of the dating problematic, an overview of the method development at PSI over the last 10 years is given in the introduction. This is followed by a description of the actual analytic procedure and comparison with the previous method is presented. Dating uncertainties are estimated, followed by an overview of different validation attempts for the 14C dating method performed by the authors group over the last years. Finally, a potential application of the method is provided by compilation of 14C dated non polar ice cores in which the discrete 14C ages are aligned with basic glaciological or empirical fits to retrieve continuous age depth relationships for the respective ice cores.

The application of the 14C method is an important topic within the challenge of ice core

dating and it is suitable to be published in The Cryosphere. The scientific level of the manuscript is already good, however I would have some suggestions and comments, which the authors might consider to further improve the quality of the manuscript.

The manuscript seems to slightly hover between a historical review and very specific descriptions of recent methodological improvements and validation exercises done by the PSI group. Since the manuscript is strongly focused on the activities of the authors group, I think the title can be somewhat misleading and I would suggest to adjust it in an adequate way. If the manuscript should keep the general review character of the 14C ice core dating topic, as the title implies until now, it could benefit from a slightly broader view and discussion of radiocarbon dating methods for glacier ice that have been developed by other groups in the past. This could for example include dating of the dissolved organic carbon fraction but also a more detailed discussion on potential dating difficulties e.g. due to reservoir effects introduced by dating of material which was already aged at deposition etc. A few references to the work of other groups would be very helpful in this case.

Specific comments:

The two-parameter model (Section 6) is only a fit to the data by adaption of the two variables, no independently derived age-depth-relationship and should possibly also be discussed like this. Nye's flow model can only be reasonably applied in the vicinity of ice divides of the polar ice sheets, where no additional horizontal flow component e.g. by a tilted bedrock, occurs. For cold Alpine glaciers frozen to bedrock it will systematically underestimate the age in larger depths. Also the assumption of a net accumulation to be constant in space and time will not hold for most Alpine glaciers. Therefore the fitted results will only give a very rough estimate of the age depth-relationship. Could you comment on that in the manuscript?

Fig. 1: I think this figure is highly redundant with the data shown in Tab. 2 and provides no additional information. The probability distributions are shown in very small size and

[Figure]

do not show any special wiggle features of the calibration curve, so in my opinion this figure could also be omitted.

Fig. 3 and Section 5: I think it is critical to infer general conclusions on the accuracy of the method for any application from only seven data points shown in this figure. It is not a priori given that this validation will work in the same way for ice bodies in all kinds of different environments and glaciological settings. I think it is good to state the successful applications discussed in this manuscript, but I would be careful to draw general conclusions for future applications of all kinds. I recommend to slightly weaken the paragraph in this respect.

Fig 6: This figure seems to be problematic to me. What is the goal to show all the ice core chronologies on one timescale? Is it just a methodological compilation of all dating applications so far? Then in my opinion it should also be discussed in a purely method-ological sense. At the moment, the figure is a compilation of ice core chronologies from glaciers with very different glaciological and geographical settings. It could be highly misleading in a fact that all the very different ice bodies are assumed to show compara-ble age-depth-relationships, which is not the case as also mentioned in the text. Could it be a possibility to compare only the glaciers with common features in one plot? Like only the cold ones or the three that have been fitted with the two-parameter model? Or separated geographically? In this context I have another question: What is the reason for the different basal behavior of the Illimani and Tsambagarav ice cores compared to the others? The basal chronology sections of these two cores differ significantly from the others. Thus, I think presenting all chronologies in one figure with only a very short glaciological description is insufficient. Either the chronologies should be discussed in a more methodological sense or the glaciological description and evaluation of the 14C-Data (section 6) needs to be extended with respect to the specific dating problems (e.g. ambiguity of volcanic horizons for the Alpine cores) of each sampling site.

Tab. 2: The radiocarbon ages derived by the Sunset method seem to be systematically higher than the ages derived by the THEODORE system. What is the reason for that,

[Figure]

can you comment? Like Referee #2, I also noticed the discrepancy between the F14C value of Bel2_THEODORE and the given radiocarbon age. I calculate an uncalibrated 14C age of 6847 yBP for that sample, thus younger than the comparable Sunset-sample. Please check.

L 11: Please specify what you mean by upper part of ice cores. How is the upper part characterised, which features separate it from the lower part?

L 16: Note that Steier et al. 2006 also published a method for micro POC 14C-dating of glacier ice. Could you give a reference here?

L 30: I think the statement about the potential of the method for dating every piece of ice is a little too general. Can you specify the prerequisites for application of the method, like concentrations of organic carbon and knowledge about the sources of the organic material?

L 41f: Could you give references for the seasonal variations of the trace components e.g. Preunkert et al. 2000 for ammonium and mineral dust components?

L 45: Change nuclear to radiometric.

L 50: Please also reference Wagenbach et al. 2012 for layer thinning and the non-linear age depth relationship of Alpine glaciers.

L 75f: Can you be a little more precise here? Not only enough carbon mass is needed, but its needs to be ensured that the dated material represents the age of the surrounding ice and was not already aged at the time of deposition. Please add a comment on that.

L 78: DOC can be extracted not only by wet oxidation but also UV-irradiation like it was done in May et al. 2013.

L 85f: Is it possible to give a reference for the fact that OC has a lower probability for built in reservoir ages than EC?

L 109: How was the stainless steel band saw pre-cleaned?

L 121: Why does the HCL has to be removed by additional rinsing with ultra-pure water?

L 128: Typo: Of

L 142f: I think the statement on graphite target measurements becoming "needless" is maybe a little too harsh. Gas measurements can only complement graphite measurements in cases where not enough material is available at the price of lower precision. Please reformulate.

L 157: Can you describe or give a reference for the protocol Swiss 4S?

L 159f: Did you take into account potential loss of material by the additional rinsing step? Did you investigate the filtration efficiency of the system in general before and after introduction of this step? This could provide information about the characteristics (size, etc.) of the retained particles. What kind of standard materials did you use? Have they been treated exactly like the ice samples?

L 201: Could you move the description of the procedural blanks from line 215 to this section? Here the term "procedural blank" occurs for the first time.

L 202: Do the given masses and F14C values only refer to the OC fraction or the total blank contribution of the system on all carbon fractions? How have the results been corrected for these blank values? Please clarify.

L 207: Not only the size but also the age has a large influence on the counting statistics and the uncertainty. Please amend.

L 210: You state that "solid grains" of the standard materials have been used. How have they been combusted? In the Sunset system or in the standard AMS preparation routine using an EA? Could you please clarify?

L 220: Why do you merge both procedural blanks of the THEODORE and the Sunset

system into one number? I think this is not reasonable, because each sample should be corrected with the blank values of the respective system. Could you explain?

L 270f: Can you give a glaciological scenario which could explain such a large age increase (ca.1000 years) in only a few cm of depth increase (below the plant fragment layer) in such a small scale, low altitude and probably temperate ice body? Do you have any information on ice temperature? Could the samples JUV0_3-JUV0_8 also be influenced by basal sediment and thus produce a significantly higher age? Please comment on that, I think only a slightly larger depth in this order of magnitude is not a sufficient explanation for the observed age increase.

L 288f: How long in depth were the subsamples? Can you provide information on the (estimated) annual layer thickness in the respective core depths and thus the expected time span covered by the subsamples? This could help to evaluate if the assumption of same 14C-age of the adjacent samples is realistic for the two upper sampling depths. In turn, can you asses if the large age increase in the basal section is realistic in terms of covered core depth of the samples? Also the grey symbols in Fig. 5 are very small and hard to distinguish from the background.

L 325: Note that especially for sites in the European Alps volcanic eruptions can be masked significantly by frequent inputs of Saharan dust (see e.g. Preunkert & Legrand 2001) and thus the signal can be highly ambiguous. Could you please comment on that?

L 335: What do you mean by "purely empirical approach"? Please clarify. Can you quantify that approach?

L 341: See comment to L 325. Please differentiate the different types of absolute horizons and their respective uncertainty.

L 349: Can you be more precise here? What are the exact prerequisites for ice bodies to be dated by the method? Is it also applicable for temperate ice, where meltwater is

present?

L 354: Please add a reference to Wagenbach et al. 2012 for the bedrock d18O-anomaly. L 357: Because of the low depth resolution, the fact of mixed ages contained in one 14C-sample holds for almost every core depth (depending on accumulation), not only for the basal layer. Please clarify.

L 379: In section 2 you stated that in total 600-800g of ice are needed for decontamination. I think this number should also be given here.

---

## Author Comment (AC1) · 26 Oct 2016

We decided to include another author, Michael Sigl, who is part of our group at PSI and discovered the fly found in the Tsambagarav ice core which was used as one of the significant validation cases. Therefore we think it is important to include him as co-author. Moreover, we acknowledge the people who prepared the sample and took the photo of the fly.

In the reviewers responses all Line numbering always refers to the revised version (track changes marked).

---

## Author Comment (AC2) · 26 Oct 2016

The authors present an overview of existing radiocarbon dating results from a suite of global ice cores, and new information on updated analysis techniques and validation exercises. On the whole, these approaches are invaluable for providing chronological information from deep portions of alpine cores, where other techniques fail yet there is still sufficient [14]C available to provide reliable dates. I have no comments or concerns regarding the updated techniques or validation procedures demonstrated in the paper - all show the care and precision typical of this group. My main comment for the authors to consider during revision is the overall presentation (and title) of the paper itself. As suggested by the title, and in the introduction (lines 90-106), on one hand the authors see this as an overview paper. But a second goal (lines 105-106) is to present recent optimizations in the analytical technique. These are not mutually exclusive; however it seems to me that the original portion of this paper is the validation exercise (section 5.2). Make no mistake, I find the optimization and validation to be novel, useful, and certainly worthy of publication here. I'm simply suggesting that the authors refine the title and introduction to reflect this. The overview portions can stay as is - The summary in section 6 and Figure 6 are quite useful for projecting how the technique could be useful in other situations.

*Response of the authors*
*Thank you very much for your review and your comments.*

*We agree that the title can be more explanatory of the content of the paper and modified it accordingly:*
***"Radiocarbon dating of glacier ice: overview, optimizations, validation and potential".***

*We also modified the last sentence of the introduction section to clarify the intention and content of the paper as following (line 109-111):*
*"Here we give an overview of the current status of the now routinely applied [14]C dating method for glacier ice including an update on recent optimizations and method validation. Uncertainties and potential of this novel approach are discussed and its successful application to a number of ice cores presented."*

---

## Author Comment (AC3) · 26 Oct 2016

*Response of the authors*
*Thank you very much for your review and your comments.*

Radiocarbon dating of glacier ice is an important asset to ice core sciences in mid-latitudes when classical methods to derive age-depth models (layer counting, ice flow modelling, tephra-chronology) fail. However, suitable material for radiocarbon dating of macrofossils is sparse in the ice making it desirable to date other organic carbon compounds. Water-insoluble organic carbon (WIOC) has been shown to be a suitable candidate for radiocarbon dating of mid-latitude ice. Uglietti et al. review the efforts to develop the analytical methodology for radiocarbon dating of WIOC in ice, and test its accuracy. The group of Bern/PSI has been instrumental in developing the method so it seems only natural that they provide a review thereof and is certainly a valuable contribution to the literature of this topic. The paper is generally well written, thorough, suitable for the Cryosphere, and should be published. However, I have some comments that the authors might want to consider for the sake of clarity in the paper and some details the authors may want to check before publication.

1. My main comment is that in the context of analytical precision, I find it slightly misleading to discuss calibrated ages instead of $^{14}$C ages or fraction modern. The calibration of radiocarbon dates is a second methodology, introducing additional uncertainty due to the uncertainty of the calibration curve itself. Hence, 2 significantly different radiocarbon ages may lead to statistically indistinguishable calibrated ages. However, if the radiocarbon dates and their uncertainties are reliable, then 2 different radiocarbon ages are indicative of different calendar ages. This applies for example to Table 2 and 3 and Figure 1. These experiments refer to analytical accuracy of radiocarbon preparation and measurement, independent of the calibrated age and it is hence worth discussing (and showing) the differences in $^{14}$C years or F$^{14}$C instead of calibrated years. A comparison of the $^{14}$C measurement results could for example be indicative whether the pure analytical precision does reflect the true uncertainty of the method. I would assume that the true uncertainty is somewhat larger, due to the inhomogeneous distribution of WIOC over an ice sample. This can for example be seen in Table 3 where the samples JUV 0_5/6 as well as JUV 0_7/8 yield significantly different radiocarbon ages, despite being from the same ice block. This questions whether these samples can be summarized to an error weighted mean and standard error as done in table 3. The final uncertainty of 9 $^{14}$C yrs. for JUV 0_3-8 seems very small given the scatter of the individual measurements. A reduced Chi2 statistic of the sample pools in Table 3 could be used to assess and discuss whether the uncertainties of single measurements are realistic. In a second step, it can then be discussed whether these uncertainties matter in terms of the absolute chronology, given that the calibration adds additional uncertainty.

*We understand that the final uncertainties of the calibrated ages are different than those of the $^{14}$C ages and also include the additional (method unrelated) uncertainty of the calibration curve. Nevertheless the main purpose of ice core dating is to provide a final age range and therefore we consider it more important to discuss and compare the uncertainties on the calibrated ages. In any case, Tables 2 to 4 do contain $^{14}$C ages and F$^{14}$C, so the information about the method related (analytical) precision is also available (also see cited literature).*
*We further agree that the uncertainty of a single measurement is indicative for the analytical uncertainty only and does not consider variation due to potential inhomogeneity in the ice sample (as indicated by different WIOC concentrations). Therefore, even if replicate results for the same sample do slightly differ from each other, considering the analytical 1σ uncertainty we think they can still be combined (in case the thinning does not suggest otherwise which of course has to be checked). In*

*fact, replicate measurements or measurements in high spatial resolution are thus preferable to avoid dating bias due to these (small) variations caused by inhomogeneity (averaging out of variations). This is feasible and a strength of the method described. After thorough consideration we however concluded that the OxCal combine tool is not appropriate to average such samples because it is not intended to take such possibility of inhomogeneity into account. We therefore revised our results by using the averages of the $^{14}C$ ages (or equal $F^{14}C$ values) with the uncertainties estimated using the standard error of the unbiased standard deviation (i.e. accounting for number of replicates) (Tables 2 and 3). Results of combined samples thus changed slightly but changes are negligible considering the uncertainties which now increased to likely more realistic estimates. We would like to emphasize, that for all the age-depth modeling performed in the past and summarized here, the model uncertainty estimates have always been selected very conservatively, one reason being exactly the consideration of potential inhomogeneity in the ice.*

*The unreasonably small uncertainty you mentioned (Table 3) was a typo. Thank you for spotting it. For JUV 0-B the standard deviation was 79 instead of the given value of 9 (it now increased to 168 years using the new approach). There was another typo: for JUV 0_7 the $^{14}C$ age uncertainty was indicated with 18 instead of 218 years (now 219, because we now consider 3 instead of 2 digits for $F^{14}C$). Using 3 digits now generally resulted in small changes of the results (usually in the order of 1 to 5 years).*

2. Please check the data in the Tables. I was confused seeing that the samples Bel2_THEODORE and Bel2_Sunset yield significantly different $F^{14}C$ values while their $^{14}C$ ages agree. Using an $F^{14}C$ of 0.425 for Bel2_THEODORE I obtain a radiocarbon age of 6874 $^{14}C$ BP, as compared to 7329 $^{14}C$ BP given in Table 2. So unless I missed something either the $F^{14}C$ or the $^{14}C$ age of this sample is erroneous which might also impact on the calibrated ages shown in Figure 1. Please check.

*We apologize; this was indeed an error. Thank you for spotting this. The Bel2_THEODORE $F^{14}C$ is 0.402, but was given as 0.425 which was picked from the wrong column in the data files and is the value before procedural blank subtraction. The correct value is given now (Table 2) and all other data in the manuscript have been cross-checked for correctness. In addition the Sunset values have slightly changed because of the blank correction. For the THEODORE – Sunset comparison the samples measured with the Sunset system were intended to be blank corrected with the corresponding blank values (1.21 ± 0.51 μg of carbon with an $F^{14}C$ of 0.73± 0.13)) but in the excel file there was an automatic link to the new combined procedural blank value (1.34 ± 0.62 μg of carbon with an $F^{14}C$ of 0.69± 0.13) which is used for all the other samples, but not intended for the comparison, as also already stated in the main text (Lines 223-225). Therefore in the new version the values appear slightly different because we used the correct blank values (1.21 ± 0.51 μg of carbon with an $F^{14}C$ of 0.73± 0.13). Moreover, we changed the sample names in Table 2 and Figure 1, to be consistent with Table 3 for the Juvfonne samples. For the Belukha samples we also changed the names. For example Bel2_THEODORE is now 4_THEODORE (BEL 2).*

3. Throughout the manuscript the term "conventional" $^{14}C$ dating is used to describe the dating of macrofossils. However, in the radiocarbon literature "conventional" $^{14}C$ dating refers to $^{14}C$ measurements using liquid scintillation and gas proportional counting techniques as opposed to AMS measurements. Please either use a different term than "conventional" or add a sentence defining how it is used in this paper.

*Yes, good point. We included a sentence to define this term in the introduction at Lines 60-61:*
*"In the following we refer to dating of ice with macrofossils as conventional $^{14}C$ dating".*

L 45 *(now line 48)*: please replace "nuclear" with "radiometric"
*We normally use the term "nuclear dating", which is common in the radiochemistry community (see for example textbook Nuclear- and Radiochemistry Vol. 2 (https://www.degruyter.com/view/product/41711).*

L 179-182: This is a very long sentence and a little unclear. Maybe divide it up into 2 sentences. Are the sample background corrected using OxII? I suppose the standard is used for normalization and not background correction? Please rephrase.

*We agree that the sentence is quite complicated and long but we think it is better to give only a short and easy explanation. We tried to improve the phrasing though (Lines 189-191):*
*"With the current setup, the $^{14}C/^{12}C$ ratio of the samples is background subtracted, normalized and corrected for mass fractionation by using fossil sodium acetate ($^{14}C$ free, NaOAc, p.a., Merck, Germany), the reference material NIST standard oxalic acid II (modern, SRM 4990C) and the $\delta^{13}C$ simultaneously measured in the AMS, respectively (Wacker et al., 2010)."*

L 183: Please insert "relative" before deviation, as the samples are normalized to the standard.

*We realized that the term "deviation" is not correct in this context and therefore confusing. We rephrase as the following (Lines 192-193):*
*"…which is the $^{14}C/^{12}C$ ratio of the sample divided by the same ratio of the modern standard"*

L 183: "BP" is not explained at this point yet, but only in line 185. Please explain it here instead.

*We corrected (Line 196):*
*"$^{14}C$ ages (before present (BP), i.e. before 1950)) are calibrated using OxCal v4.2.4 (Bronk Ramsey and Lee, 2013) with the Northern (IntCal13) or Southern Hemisphere (ShCal13) calibration curves (Reimer et al. 2013, Hogg et al. 2013), depending on the sample site location. Calibrated dates are given in years before present (cal BP)".*

L 227-228: Are the uncertainties given in $^{14}C$ years here? If so, please write "$^{14}C$ yrs" instead of years to be clear.

*We here talk about an assumed age of the ice sample and thus refer here to the "true" age and not the $^{14}C$ age. Accordingly the uncertainties denote the final overall uncertainty of the dating method which as such also includes the calibration so they refer to the calibrated age. To clarify we changed the sentence to "As an example, for hypothetical samples with a WIOC mass of 5 or 10 µg, the resulting uncertainty of the finally calibrated ages for 1000 year old ice would be around ± 600 yrs or ± 250 yrs and for 8000 year old ice around ± 1600 yrs or ± 700 yrs, respectively (Line 249-253)*

L 249 and following: See comment number 3. Please either define what you mean by conventional or use a different term throughout the manuscript instead.

*See reply above (comment 3.) and see Lines 60-61 in revised version.*

L 252: This may be nitpicking, but AD 1258 – AD 1050 = 208 years, not 200 years.

*With an error of 70 years, rounding to down to 200 seems appropriate. However, we changed to "around 200 years" then omitting the need for indication of the uncertainty which could be derived anyhow with the value of AD 1050±70 being provided (Line 274).*

L 259-271: Several times it is stated that the WIOC dates "agree well" with the macrofossil dates, while the $^{14}C$ ages are indeed significantly different. I am not arguing against the general agreement but it would be great if you could add 1-2 sentences to make this more precise. Are the differences due to sampling differences (i.e., different ice layers have been sampled for macrofossils and WIOC)? If so, are the results in stratigraphic order?

*Yes, the differences arise from the sampling procedure. The samples in the Juvfonne ice patch were obtained extracting clear ice just adjacent to the organic remains layers where the macrofossil $^{14}C$ ages are from. Thus a 1:1 agreement cannot be expected. Here, the samples from Ødegard et al. (2016) are simply given as a reference age range to compare our samples with. To allow for the most reasonable comparison we always considered the organic layers closest to our ice layers. In this*

*version we thus changed for example the organic layer Poz-37879 with Poz-37877 which is closer to the ice sample JUV 2 and was not available for the original study by Zapf et al., 2013 which is why we did not consider it initially but now decided to. The same for JUV 1 and the corresponding closest organic layers Poz-36460 which we kept and Poz-37878 which was replaced with a closer layer Poz-56952 only sampled in 2012 and therefore also not available in the study by Zapf et al. 2013.*

*We now describe the issue more carefully by including the stratigraphic order in Table 3 and an explanation in the related caption text and also in the main text (Lines 292-294).*

L 351: Please write "climate wiggle matching" instead of just "wiggle matching" which could also refer to the wiggle matching of radiocarbon dates.

*Yes, done, thanks (Line 395).*

L 357: Please add a reference to [Godwin, H. 1962. Nature, 195 (4845)] for the half-life of radiocarbon.

*Yes added. Thank you (Line 402).*

---

## Author Comment (AC4) · 26 Oct 2016

*Response of the authors*
*Thank you very much for your review and your comments.*

The manuscript "Radiocarbon dating of glacier ice" by Uglietti et al. aims to give an overview of the actual 14C dating method applied by the authors group (i.e. at the Paul-Scherrer-Institute (PSI) for absolute dating of glacier ice via 14C analysis of the water insoluble organic carbon fraction WIOC. After the presentation of the dating problematic, an overview of the method development at PSI over the last 10 years is given in the introduction. This is followed by a description of the actual analytic procedure and comparison with the previous method is presented. Dating uncertainties are estimated, followed by an overview of different validation attempts for the 14C dating method performed by the authors group over the last years. Finally, a potential application of the method is provided by compilation of 14C dated non polar ice cores in which the discrete 14C ages are aligned with basic glaciological or empirical fits to retrieve continuous age depth relationships for the respective ice cores.

The application of the 14C method is an important topic within the challenge of ice core dating and it is suitable to be published in The Cryosphere. The scientific level of the manuscript is already good, however I would have some suggestions and comments, which the authors might consider to further improve the quality of the manuscript.

The manuscript seems to slightly hover between a historical review and very specific descriptions of recent methodological improvements and validation exercises done by the PSI group. Since the manuscript is strongly focused on the activities of the authors group, I think the title can be somewhat misleading and I would suggest to adjust it in an adequate way. If the manuscript should keep the general review character of the [14]C ice core dating topic, as the title implies until now, it could benefit from a slightly broader view and discussion of radiocarbon dating methods for glacier ice that have been developed by other groups in the past. This could for example include dating of the dissolved organic carbon fraction but also a more detailed discussion on potential dating difficulties e.g. due to reservoir effects introduced by dating of material which was already aged at deposition etc. A few references to the work of other groups would be very helpful in this case.

*We revised the title as the following as also suggested by other reviewers: "**Radiocarbon dating of glacier ice: overview, optimizations, validation and potential**".*

*We do give references to other groups regarding [14]C in DOC (see Introduction). These studies include method validation but no published DOC ice core dating results which is the topic of this overview. For studies of [14]C in POC, see comment further down. Concerning the potential dating difficulties (e.g. reservoir effect) we discuss it already in Lines 84 to 89 (**line numbering related to revised version**). We did include a sentence though (Lines 327-328), that samples of high dust load were avoided giving also a reference to Hoffmann, 2016 (also see comment further down).*

**Specific comments:**
The two-parameter model (Section 6) is only a fit to the data by adaption of the two variables, no independently derived age-depth-relationship and should possibly also be discussed like this. Nye's flow model can only be reasonably applied in the vicinity of ice divides of the polar ice sheets, where no additional horizontal flow component e.g. by a tilted bedrock, occurs. For cold Alpine glaciers frozen to bedrock it will systematically underestimate the age in larger depths. Also the assumption of a net accumulation to be constant in space and time will not hold for most Alpine glaciers. Therefore the fitted results will only give a very rough estimate of the age depth-relationship. Could you comment on that in the manuscript?

*We completely agree that Nye's flow model cannot be applied for all the reasons stated by the reviewer. This is why absolute dating of the bottom ice is so important especially for Alpine glaciers.*

*This was in fact one of the main drivers to develop this method since also much more complex flow models are not able to give reliable results for this part of the cores.*

*The two parameter model (Thompson et al 1990) is exactly used just as a fit through the reference horizons and the $^{14}C$ data points to derive a best estimate of a continuous age-depth relationship (see Line 339). Considering the size of the $^{14}C$ uncertainties it should of course only be "rough" but does allow an estimate at least. In any case, there is no intent of using the model to infer absolute dates or confirm the dating results as clearly the obtained age ranges are derived based on the $^{14}C$ analyses (and the other time horizons in the younger part).*

*However, being unable to fit the absolute time markers with the two parameters model (e.g. Ortles and Tsambagarav ice cores) can give some indication that: (i) the underlying assumptions of constant accumulation does not hold and net accumulation might have changed significantly over time which is of course a possibility and should then be investigated more carefully (e.g. Tsambagarav and likely also Illimani although there the uncertainties might be too large for a solid conclusion, find more details in the cited original references for both) or (ii) the degree of layer thinning might be different from what the rather simple model can assume with the given degree of freedom to the parameters (e.g. Ortles, see cited reference Gabrielli et al., 2016). In the revised version we modified the relevant paragraph in order to clarify (Lines 362-386).*

Fig. 1: I think this figure is highly redundant with the data shown in Tab. 2 and provides no additional information. The probability distributions are shown in very small size and do not show any special wiggle features of the calibration curve, so in my opinion this figure could also be omitted.

*We do not agree and we consider the figure self-explicative and significant to be shown, while the table is needed to document the data we used to produce the figure.*

*The probability distributions are compressed and therefore less clear only for the modern samples (which are not really relevant for the dating by $^{14}C$ anyhow). Anyway, they are shown here as another piece of evidence that the comparison between the two combustion systems is good.*

Fig. 3 and Section 5: I think it is critical to infer general conclusions on the accuracy of the method for any application from only seven data points shown in this figure. It is not a priori given that this validation will work in the same way for ice bodies in all kinds of different environments and glaciological settings. I think it is good to state the successful applications discussed in this manuscript, but I would be careful to draw general conclusions for future applications of all kinds. I recommend to slightly weakening the paragraph in this respect.

*We do not think one can reduce the presented results to "only seven data points". First of all there is also strong evidence for the validity of the method based on the larch leaf in the Ortles core which is not shown in this figure and second, these 8 (including Ortles) points do represent 4 different sites from very different regions, different environments and different glaciological settings and thus can be considered to be rather representative. Anyhow, we now added a sentence saying that all validation experiments were performed on low-dust samples, thus avoiding potential issues related to high dust content in the dating (Hoffman 2016). Lines 327-328.*

Fig 6: This figure seems to be problematic to me. What is the goal to show all the ice core chronologies on one timescale? Is it just a methodological compilation of all dating applications so far?

*Yes, indeed the figure is exactly a compilation of all dating applications to show that for these ice cores it was the only method to date ice older than 1000 years.*

*The figure also summarizes (better than words on our view) typical ages, depths and thinning rates for high-alpine ice cores which are very different to those from polar ice cores.*

Then in my opinion it should also be discussed in a purely methodological sense. At the moment, the figure is a compilation of ice core chronologies from glaciers with very different glaciological and

geographical settings. It could be highly misleading in a fact that all the very different ice bodies are assumed to show comparable age-depth-relationships, which is not the case as also mentioned in the text.

*Yes, the text clearly states that we do not assume that all the ice cores have the same age-depth relationship. To clarify this more clearly, we changed the first sentence in the figure caption to: "Compilation of age-depth relationships for five different ice cores, highlighting the importance of the WIOC 14C dating to obtain continuous chronologies and to constrain the very specific glaciological conditions and settings of each site.."*

Could it be a possibility to compare only the glaciers with common features in one plot? Like only the cold ones or the three that have been fitted with the two-parameter model? Or separated geographically? In this context I have another question: What is the reason for the different basal behavior of the Illimani and Tsambagarav ice cores compared to the others? The basal chronology sections of these two cores differ significantly from the others. Thus, I think presenting all chronologies in one figure with only a very short glaciological description is insufficient. Either the chronologies should be discussed in a more methodological sense or the glaciological description and evaluation of the 14C-Data (section 6) needs to be extended with respect to the specific dating problems (e.g. ambiguity of volcanic horizons for the Alpine cores) of each sampling site.

*This is not intended to be a glaciological paper. The intention of this figure is to show the various applications of the method for the dating of ice cores previously presented in much greater detail elsewhere (see cited references). There, more specific glaciological descriptions, careful considerations regarding the time horizons selected (e.g. volcanic horizons) and the interpretation of the age scale and analyzed parameters in general can be found.*

*Regarding the specific questions about the basal ages of Illimani and Tsambagarav we also would like to refer to the original literature for more details although for Tsambagarav it is already discussed in the manuscript here (significant change in accumulation, see Herren et al. 2013).*

*To clarify this, the paragraph has been modified and slightly extended (lines 362-383 in the revised version). Please also see further up where related and additional explanation is provided (first paragraph of the specific comments related to the two parameter model).*

Tab. 2: The radiocarbon ages derived by the Sunset method seem to be systematically higher than the ages derived by the THEODORE system. What is the reason for that, can you comment? Like Referee #2, I also noticed the discrepancy between the $F^{14}C$ value of Bel2_THEODORE and the given radiocarbon age. I calculate an uncalibrated 14C age of 6847 yBP for that sample, thus younger than the comparable Sunset sample.
Please check.

*Thank you for spotting the erroneous $F^{14}C$ value of the Bel2_THEODORE sample (see answer to Referee #2). Three of the samples give ages equal considering 1sigma for both systems (1, 2, 4), whereas only for sample 3 the Sunset results in older ages. Taking into account that these samples are not aliquots and may show some inhomogeneity, we think there is not sufficient evidence for a systematic bias.*

L 11: Please specify what you mean by upper part of ice cores. How is the upper part characterized, which features separate it from the lower part?
*Upper is related to the first 200 years which can be dated with the Pb210 method. In term of depth, this varies for each ice core dependent on the accumulation rate.*

L 16: Note that Steier et al. 2006 also published a method for micro POC 14C-dating of glacier ice. Could you give a reference here?
*We do not like to add a reference in the abstract, but we do cite Steier et al. 2006 at line 87.*

L 30: I think the statement about the potential of the method for dating every piece of ice is a little too general. Can you specify the prerequisites for application of the method, like concentrations of organic carbon and knowledge about the sources of the organic material?

*We already specified the needed OC concentration and the approximate size (mass) of the ice piece needed (see lines 18 to 23 in the revised version). The purely biogenic source of WIOC prior to industrialization is discussed later in the introduction (Lines 85-93) with reference to the according study from Jenk et al., 2006.*

L 41: Could you give references for the seasonal variations of the trace components e.g. Preunkert et al. 2000 for ammonium and mineral dust components?

*Yes, it has been included (line 43 in revised version).*

L 45: Change nuclear to radiometric.

*We normally use the term "nuclear dating", which is common in the radiochemistry community (see for example textbook Nuclear- and Radiochemistry Vol. 2 (https://www.degruyter.com/view/product/41711).*

L 50: Please also reference Wagenbach et al. 2012 for layer thinning and the non-linear age depth relationship of Alpine glaciers.

*In Wagenbach et al. 2012 no age-depth relationship and corresponding layer thinning of the deeper part is presented. Instead the "basal layer enigma" is evoked. We therefore think it is not relevant to refer to that publication in this section of our manuscript. We refer to that paper in section 6.*

L 75: Can you be a little more precise here? Not only enough carbon mass is needed, but its needs to be ensured that the dated material represents the age of the surrounding ice and was not already aged at the time of deposition. Please add a comment on that.

*It was specified a few lines later in the introduction (Lines 84 to 93).*

L 78: DOC can be extracted not only by wet oxidation but also UV-irradiation like it was done in May et al. 2013.

*Also the oxidation by UV-irradiation takes place in the wet phase. We therefore consider "wet oxidation" sufficient.*

L 85: Is it possible to give a reference for the fact that OC has a lower probability for built in reservoir ages than EC?

*Thank you, yes, Sigl et al 2009 and Gavin et al 2001 (Line 91)*

L 109: How was the stainless steel band saw pre-cleaned?

*The Band Saw Blades are cleaned before use with acetone on paper towels until the towels come out white (usually 3 times). A MQ frozen blank is also cut prior to samples cutting. We include this information in the description of the method (Lines 116-117).*

L 121: Why does the HCL has to be removed by additional rinsing with ultra-pure water?

*It should be removed in order not to interfere with or even damage the IR detector in the Sunset instrument as it was already written in Lines 168-169.*

L 128: Typo: Of

*Thanks, but the upper case O is intentional because of the acronym: **T**wo-step **H**eating system for the **EC/OC D**etermination **Of R**adiocarbon in the **E**nvironment (**THEODORE**, Szidat et al., 2004)*

L 142: I think the statement on graphite target measurements becoming "needless" is maybe a little too harsh. Gas measurements can only complement graphite measurements in cases where not enough material is available at the price of lower precision.
Please reformulate.

*We do not agree with this comment of the reviewer. Given the context of this sentence, we do not state that graphitization is needless in general. We emphasize that graphitization became needless for us and we describe how we were able to overcome this additional step. Anyhow, for small samples sizes (<100 µgC), uncertainties of gas and graphite measurements are similar. For larger sizes, graphite formation is still favorable for better dating precisions. However, all samples discussed in this manuscript are far below 100 µgC so that we regard further comments as not necessary in this context.*

L 157: Can you describe or give a reference for the protocol Swiss 4S?
*The reference was already given at the end of the sentence; it is Zhang et al 2012 (Line 168).*

L 159: Did you take into account potential loss of material by the additional rinsing step? Did you investigate the filtration efficiency of the system in general before and after introduction of this step? This could provide information about the characteristics (size, etc.) of the retained particles. What kind of standard materials did you use?
Have they been treated exactly like the ice samples?

*The rinsing step involves only 5 ml of MQ water compared to the ca 500 ml of sample (melted ice containing particles) which is flowing through the glass funnel and the filter during the filtration process. There is a potential loss of small particles on the walls of the glass funnel and through the filter (we are limited to the quartz fiber filters), but neither the entity of the loss nor the size of the particles is known. Nevertheless we are quite confident that the extra rinsing step does not significantly influence the filtration efficiency, again because the rinsing is done with only few ml of water.*

*We decided to remove the sentence regarding the tests with blanks and standard materials as it does not really fit here considering the context of this paragraph (removed from line 164 in the reviewed version). The current set-up and all set-ups used in the past (THEODORE, uncoupled Sunset system) have been thoroughly tested for the blank of the instrumental part (i.e. system blank). In this context, the additional step of rinsing after acidification was tested and was found to not add additional contamination. The entire ice samples preparation procedure results in the overall procedural blank described elsewhere in the manuscript. It includes this additional step, therefore the information will still be contained even if the sentence is deleted. The tests using the standard materials have also been repeatedly performed for each of the setups (e.g. Jenk et al., 2007; e.g. Zhang et al., 2012; and Agrios et al., 2015 respectively). Details about the standard materials used (IAEA and NIST standards such as IAEA-C5, C6, C7 or NISTHOx2) can be found in these studies which are cited in the manuscript. In order not to lose focus on the main point of this review paper due to an overload with technical details we think they should not be listed here again.*

*No, the treatment of standards was not exactly like the ice samples but the ice blanks were (resulting in the procedural blank values). The standards can ensure precision and accuracy of the instrumental part only. It is highly challenging to produce ice containing standard materials (introduced inhomogeneity due to the freezing step, loss to container walls etc.) which is a general issue for most parameters analyzed in ice cores. For this study, this is one of the main reasons and strong motivation for the discussed method validation experiments with independently dated ice.*

L 201: Could you move the description of the procedural blanks from line 215 to this section? Here the term "procedural blank" occurs for the first time.

*Thank you for spotting it. We prefer not to change the structure of the manuscript because the current sequence makes the most sense to us, but we now included a reference to the next Section 4 (Line 194-195).*

L 202: Do the given masses and $F^{14}C$ values only refer to the OC fraction or the total blank contribution of the system on all carbon fractions? How have the results been corrected for these blank values? Please clarify.

*Thanks for the comment. Later in the manuscript, in Section 4, we state that procedural blanks are treated similarly to the samples (Lines 236-237). This also includes the combustion step (Line 239) which in this case is for the OC (i.e. WIOC) fraction. So, yes they do refer only to the WIOC (information added in line 240 in revised version). All results from ice samples are corrected for the procedural blank, which mean the $F^{14}C$ values are blank subtracted using an isotopic mass balance equation which can be found in the original paper where the method is presented for the first time (Jenk et al., 2007).*

L 207: Not only the size but also the age has a large influence on the counting statistics and the uncertainty. Please amend.

*We here made a first and very general statement for easy understanding of the main issue with counting statistics. In lines 248-249 we then did amend the influence of the age: "Low carbon mass samples of old age contain even a lower number of $^{14}C$ compared to younger samples due to radioactive decay and are affected the most".*

L 210: You state that "solid grains" of the standard materials have been used. How have they been combusted? In the Sunset system or in the standard AMS preparation routine using an EA? Could you please clarify?

*As for the samples, we combusted the standards in the Sunset. Although this seems to be clear to us, since we never mentioned the EA, we added this info to the text (Line 229).*

L 220: Why do you merge both procedural blanks of the THEODORE and the Sunset system into one number? I think this is not reasonable, because each sample should be corrected with the blank values of the respective system. Could you explain?

*Because the blank values over the 10 years of analyses with the THEODORE and the new Sunset values were not observed to be significantly different from each other. For both systems the pure system blank which does not include the ice sample preparation has been thoroughly investigated (needed for correction of standard material measurements etc.). We know, that by far the major contribution of the procedural blank is related to the ice sample preparation and filtration step. Both these steps are not dependent on the instrumentation (i.e. whether the THEODORE or Sunset system is used). For all these reasons and based on the data currently available a combination of all blank values in order to get the most reliable value and the best statistics (best uncertainty estimate) seems to be the most appropriate approach. We explain this in the manuscript now (Lines 241-245).*

L 270: Can you give a glaciological scenario which could explain such a large age increase (ca.1000 years) in only a few cm of depth increase (below the plant fragment layer) in such a small scale, low altitude and probably temperate ice body? Do you have any information on ice temperature? Could the samples JUV0_3-JUV0_8 also be influenced by basal sediment and thus produce a significantly higher age? Please comment on that, I think only a slightly larger depth in this order of magnitude is not a sufficient explanation for the observed age increase.

*This is an interesting question. For this ice patch, many more organic remains layers have been analyzed by the conventional $^{14}C$ method than the ones presented here (Ødegård, et al. 2016). Based on those, the large age increase is not unexpected. Possible explanations are discussed in detail in Ødegård, et al. 2016*

L 288: How long in depth were the subsamples? Can you provide information on the estimated) annual layer thickness in the respective core depths and thus the expected time span covered by the subsamples? This could help to evaluate if the assumption of same 14C-age of the adjacent samples is realistic for the two upper sampling depths. In turn, can you asses if the large age increase in the basal section is realistic in terms of covered core depth of the samples? Also the grey symbols in Fig. 5 are very small and hard to distinguish from the background.

*Regarding the samples of the two upper sampling depths and the combination of sub-samples to one age, please see comment made to point 1 of Referee #2. Also, when combining samples to a common age we do of course always consider first if this is reasonable in terms of the expected thinning (see e.g. Fig. 5 for Mt. Ortles). Related to that, please refer to Ødegård, et al. 2016 for the expected thinning in the JUV ice patch (also see previous comment).*

L 325: Note that especially for sites in the European Alps volcanic eruptions can be masked significantly by frequent inputs of Saharan dust (see e.g. Preunkert & Legrand 2001) and thus the signal can be highly ambiguous. Could you please comment on that?

*We are very well aware of the potential impact of dust on the detection of volcanic signals in ice cores. We used multiple records to identify Saharan dust layers (e.g. Fe, $Ca^{2+}$) which by themselves can be valuable isochrones (on a regional scale). We were very conservative with the attribution of volcanic events in the Alps and are confident that those glacio-chemical signals we attributed to the few large magnitude eruptions (Laki, Tambora, Novarupta) are correctly identified. Nevertheless the topic of this paper is radiocarbon dating of glacier ice and not dating of glacier ice in general. In order to keep the storyline of the manuscript straight, we omit to include this aspect here (in the cited literature more details can be found by the readers interested in this topic).*

L 335: What do you mean by "purely empirical approach"? Please clarify. Can you quantify that approach?

*We intended to make the point that the model is not based on physical flow laws.*

L 341: See comment to L 325. Please differentiate the different types of absolute horizons and their respective uncertainty.

*In order to keep the storyline of the manuscript straight, we omit to include this aspect in more detail. For the different types of the reference horizons used we would like to refer to each original paper which we cited. The uncertainties are already included for both $^{14}C$ and RH by the error bars in Figure 6 (the text in the according Figure caption has been changed to clarify). Note that they are masked by the symbol size in some cases.*

L 349: Can you be more precise here? What are the exact prerequisites for ice bodies to be dated by the method? Is it also applicable for temperate ice, where meltwater is present?

*We included the info on cold and polithermal ice bodies (added at line 394 of the revised version). Until now we did not date any temperate ice body with $^{14}C$.*

L 354: Please add a reference to Wagenbach et al. 2012 for the bedrock d$^{18}$Oanomaly.

*Thank you, we did add it.*

L 357: Because of the low depth resolution, the fact of mixed ages contained in one 14C-sample holds for almost every core depth (depending on accumulation), not only for the basal layer. Please clarify.

*We clarify it by rephrasing as the following (Lines 400-403): "Because of the strong thinning, the $^{14}C$ age of the deepest sample represents a strongly mixed age of ice with a large age distribution. In*

*these cases, the age limit was thus not determined by the $^{14}$C half-life of 5730 yrs (Godwin et al 1962) but by the achievable spatial depth resolution since some hundred grams of ice is required."*

L 379: In section 2 you stated that in total 600-800g of ice are needed for decontamination. I think this number should also be given here.
*The corresponding number was added (Lines 425-427 in the revised version) although the value of 600-800g of ice we gave in Section 2 was unfortunately but obviously incorrect (we apologize for that and corrected in this version). The ice mass required for the analysis is around 200 - 500g which is the mass of the decontaminated sample. The mass loss due to decontamination is around 20 - 30 % as stated in Section 2. With this, one can easily calculate that the number for the initial ice mass should be around 300 - 800 g of ice (conservative estimate, i.e. rounded up).*

---

## Author Comment (AC6) · 26 Oct 2016

The comment was uploaded in the form of a supplement:
http://www.the-cryosphere-discuss.net/tc-2016-160/tc-2016-160-AC6-supplement.pdf